# Lack of IL-1R8 in neurons causes hyperactivation of IL-1 receptor pathway and induces MECP2-dependent synaptic defects

Romana Tomasoni[1,2], Raffaella Morini[1], Jose P Lopez-Atalaya[2], Irene Corradini[1,3], Alice Canzi[1,4], Marco Rasile[1,4], Cristina Mantovani[1], Davide Pozzi[1,4], Cecilia Garlanda[1], Alberto Mantovani[1,4], Elisabetta Menna[1,3], Angel Barco[2], Michela Matteoli[1,3]*

[1]IRCCS Humanitas, Rozzano, Italy; [2]Instituto de Neurociencias (Universidad Miguel Hernández-Consejo Superior de Investigaciones Científicas), Alicante, Spain; [3]IN-CNR, Milano, Italy; [4]Hunimed University, Rozzano, Italy

**Abstract** Inflammation modifies risk and/or severity of a variety of brain diseases through still elusive molecular mechanisms. Here we show that hyperactivation of the interleukin 1 pathway, through either ablation of the interleukin 1 receptor 8 (IL-1R8, also known as SIGIRR or Tir8) or activation of IL-1R, leads to up-regulation of the mTOR pathway and increased levels of the epigenetic regulator MeCP2, bringing to disruption of dendritic spine morphology, synaptic plasticity and plasticity-related gene expression. Genetic correction of MeCP2 levels in IL-1R8 KO neurons rescues the synaptic defects. Pharmacological inhibition of IL-1R activation by Anakinra corrects transcriptional changes, restores MeCP2 levels and spine plasticity and ameliorates cognitive defects in IL-1R8 KO mice. By linking for the first time neuronal MeCP2, a key player in brain development, to immune activation and demonstrating that synaptic defects can be pharmacologically reversed, these data open the possibility for novel treatments of neurological diseases through the immune system modulation.

*For correspondence: michela. matteoli@humanitasresearch.it

## Introduction

Neurological disorders represent an enormous source of burden to the individual and to society, with many patients failing to respond to available medication. Growing evidence on genetic components of neurological diseases have been collected during recent years; notably, these genes overwhelmingly point to disorders of synaptic transmission, which led to the coinage of the term 'synaptopathy' to indicate a brain disease originating from a dysfunction of the synapse (*Grant, 2012*; *Grabrucker, 2014*). Disruption of synapse function may also be caused by environmental stimuli, with inflammatory cytokines affecting synaptic transmission and modifying the risk and severity of a variety of brain diseases, including autism spectrum disorders, schizophrenia and cognitive disabilities (*Hagberg et al., 2012*; *Chugh et al., 2013*; *Steinman, 2013*). Among the cytokines known to affect synaptic function, the proinflammatory cytokine IL-1$\beta$ plays a critical role. IL-1$\beta$ was found to impair brain-derived neurotrophic factor (BDNF)-induced expression of molecules critical for activity-dependent synaptic plasticity, including cAMP response element binding protein (CREB), Arc, and cofilin, thus reducing actin polymerization and impairing spine morphogenesis (*Tong et al., 2012*). In addition, IL-1$\beta$ controls different neuronal functions, including excitability and transmitter release, via multiple biochemical pathways (*Weber et al., 2010*). However, no evidence

**eLife digest** Errors that occur while the brain is developing can lead to conditions such as autism and schizophrenia. They can also lead to rare disorders like Rett syndrome and MeCP2 duplication syndromes, which are characterized by severe cognitive and physical disabilities. Many people with these neurodevelopmental disorders have mutations in genes that encode proteins found at synapses, which are the junctions between neurons where the cells exchange information with one another. However, not everyone with these mutations develops a neurodevelopmental disorder, which indicates that other, non-genetic factors also play a part.

One of the main non-genetic factors that can influence the risk and severity of neurodevelopmental disorders is inflammation of the brain. Inflammation is a normal part of the body's immune response to threats such as invading microorganisms or tissue damage. However, abnormal activation of the immune system in early life can trigger excessive inflammation. This increases the risk of a neurodevelopmental disorder, but it is not clear exactly how it does so.

Tomasoni et al. set out to test whether the missing link between inflammation and neurodevelopmental disorders might be damage to synapses. The experiments revealed that genetically modified mice with inflammation of the brain have abnormal synapses and are unable to learn properly. These mutant mice also have excessive levels of a protein that influences how synapses function called MeCP2, which is missing in the brains of people with Rett syndrome and abnormally increased in brains of patients affected by MeCP2 Duplication Syndrome. This is thus the first evidence that directly links inflammation of the brain to a synapse protein implicated in a disorder of brain development.

Tomasoni et al. also found that a drug called anakinra – which is used to treat an inflammatory disease called rheumatoid arthritis – reduced levels of MeCP2 in the mutant mice and improved their performance in cognitive tasks. Together, these results raise the possibility that anti-inflammatory medications may be beneficial in the treatment of neurodevelopment disorders.

has been reported yet that IL-1$\beta$ may affect synapse development and function acting on molecular pathways known to be at the root of synaptopathies.

Genetic mouse models of immune deregulation may serve as reliable and reproducible systems for examination of the effects of inflammation on synapse structure and function and elucidation of the molecular processes involved. IL-1R8, also known as single Ig IL-1-Related Receptor (SIGIRR) or TIR8, belongs to the toll-like receptors (TLRs) and interleukin-1R-like receptors (ILRs), a family of conserved proteins involved in immunity and inflammation (*Riva et al., 2012*; *Garlanda et al., 2013a*). TLRs are receptors able to recognize specific pathogen-associated patterns (PAMPs) and necrotic cell-derived danger signals (DAMPs) and act as sensors for microorganisms and tissue damage, whereas the IL-1R subfamily includes components of signalling receptor complexes as well as molecules with regulatory function. IL-1R8 dampens the activation of the TLRs and IL-1R signalling pathways by intracellularly interfering with the association of adaptor molecules to the receptor complexes, including NF-κB and JNK (*Riva et al., 2012*; *Garlanda et al., 2013b*). As a consequence, IL-1R8-deficient mice display exaggerated symptoms of inflammatory conditions (*Garlanda et al., 2007*; *Gulen et al., 2010*; *Drexler et al., 2010*) and demonstrate pronounced susceptibility to the inflammatory challenge posed by microbial LPS (*Garlanda et al., 2004*). IL-1R8 is also present in the brain (*Costelloe et al., 2008*; *Polentarutti et al., 2003*). Genetic deficiency for IL-1R8 is associated with inflammatory changes in the brain, including increased levels of LPS-induced tumor necrosis factor α (TNFα) and IL-6 in microglia, higher expression of TLR4, and NF-κB activation (*Watson et al., 2010*). Reduced IL-1R8 expression has been described in patients affected by psoriatic arthritis (*Batliwalla et al., 2005*) while SIGIRR variants (characterized by defective SIGIRR function) have been found in humans in association with necrotizing enterocolitis (*Sampath et al., 2015*) and with systemic lupus erythematosus (SLE; [*Zhu et al., 2014*]), all pathologies being characterized by cognitive defects and neurodevelopmental impairment (*Husted et al., 2013*; *Rees et al., 2007*; *Calderón et al., 2014*; *Muscal et al., 2010*). Notably, IL-1R8-deficient mice are impaired in novel object recognition, spatial reference memory and long-term potentiation (LTP), defects that occur in

the absence of any external inflammatory stimuli (*Costello et al., 2011*). However, the molecular mechanisms by which IL-1R8 deficiency results in brain defects are still completely unknown.

We used IL-1R8 deficient mice to investigate whether genetic hyperactivation of the IL-1R pathway affects synapse function impinging on molecular players involved in synaptopathies. We demonstrate that the activity of the IL-1R pathway directly affects, in neurons, the levels of expression of the methyl-CpG-binding protein 2 (MeCP2), a synaptopathy protein involved in neurological diseases -Rett syndrome and MeCP2 duplication syndrome- characterized by defective plasticity, impaired cognition and intellectual disability. We also show that pharmacological inhibition of IL-1R activity normalizes MeCP2 expression and cognitive deficits in IL1R8-deficient mice.

## Results

### Altered synaptic architecture and function in IL-1R8 KO hippocampal neurons

To investigate the impact of IL-1R8 deficiency on synapse structure and function, we examined the morphology and plasticity of dendritic spines in primary cultures established from embryonic IL-1R8 KO or WT mice hippocampi, transfected with GFP at DIV 12. Compared to their WT counterpart, IL-1R8 KO neurons displayed an increased number of immature, thin spines and a decreased number of mature, mushroom spines (*Figure 1A–C*) along with a significant reduction of spine width (*Figure 1D*). Also PSD-95 puncta density (*Figure 1E*) and size (*Figure 1F*) were significantly lower in IL-1R8 KO neurons. Consistently, the levels of the postsynaptic protein PSD-95, evaluated by western blotting of culture homogenates were significantly reduced (*Figure 1G and H*). In line with a synaptic defect, patch clamp recording of IL-1R8 deficient cultures revealed that the frequency, but not the amplitude, of miniature excitatory postsynaptic currents (mEPSCs) was significantly reduced (*Figure 1I–K*). Spine defects (*Figure 2A and B*) and reduction in synaptic markers (*Figure 2C and D*) were also detected in CA1 pyramidal neurons of IL1R8 KO mice with respect to age-matched WT controls.

To investigate whether IL-1R8 KO neurons are able to undergo synaptic potentiation, hippocampal cultures were subjected to an established chemical LTP (c-LTP) protocol based on the culture exposure to 100 µM glycine in KRH devoid of Mg, followed by a washout and recovery in neuronal medium for at least 60 min (*Menna et al., 2013*). Under these conditions, a significant increase in the density of both PSD-95-positive puncta and mushroom spines occurred in WT but not in IL-1R8 KO neurons (*Figure 3A–C*). Similarly, no increase in mEPSC amplitude and frequency was recorded over time in IL-1R8 KO neurons (*Figure 3D–F*), univocally indicating that hippocampal IL-1R8 KO neurons are unable to undergo synaptic plasticity. These data show the occurrence of synaptic structural and functional defects in primary cultures from IL-1R8 KO mice.

### Defects in the structure and function of IL-1R8 KO neurons are reversed by blocking IL-1 receptor activity

To determine whether the synaptic defects of IL-1R8 KO neurons are attributable to IL-1 receptor (IL-1R) or TLR pathway, both negatively regulated by IL-1R8, we analyzed spine density and electrophysiological properties in neurons from mice deficient for both IL-1R8 and IL-1R (IL-1R8 KO IL-1R KO). Double IL-1R8 KO IL-1R KO neurons displayed an increase in mushroom spine density (number of spines/micron, mean ± SEM, WT: 0,1581 ± 0,01508, n = 32; IL-1R8 KO IL-1R KO: 0,2585 ± 0,0154, n = 41, Student t test, p<0,0001), accompanied by enhanced mEPSCs frequency and amplitude (mEPSC frequency, mean ± SEM, WT: 1317 ± 0,1714, n = 12; IL-1R8 KO IL-1R KO: 2432 ± 0,3187, n = 17, Student t test, p<0,05; mEPSC amplitude, mean + SEM, WT: 22,82 ± 2468, n = 12; IL-1R8 KO IL-1R KO: 29,10 ± 2112, n = 17, Student t test, ns, p=0,0644). These data suggest a role for IL-1R signaling in controlling synaptic structure and function. To determine if reducing IL-1R activity could restore synaptic plasticity and spatial learning, hippocampal neurons from IL-1R8 KO mice exposed overnight to IL1Ra (Anakinra), a naturally-occurring IL-1 receptor antagonist (*Dinarello, 2009*), displayed increased density of both mushroom spines (*Figure 4A and B*) and PSD-95 puncta (*Figure 4A and C*). Furthermore, IL1Ra-treated neurons from IL-1R8 KO mice recovered their ability to undergo both structural and functional LTP, as indicated by the increased density of spines and PSD-95 (*Figure 4A–C*) and mEPSC frequency and amplitude (*Figure 4D–F*) following the

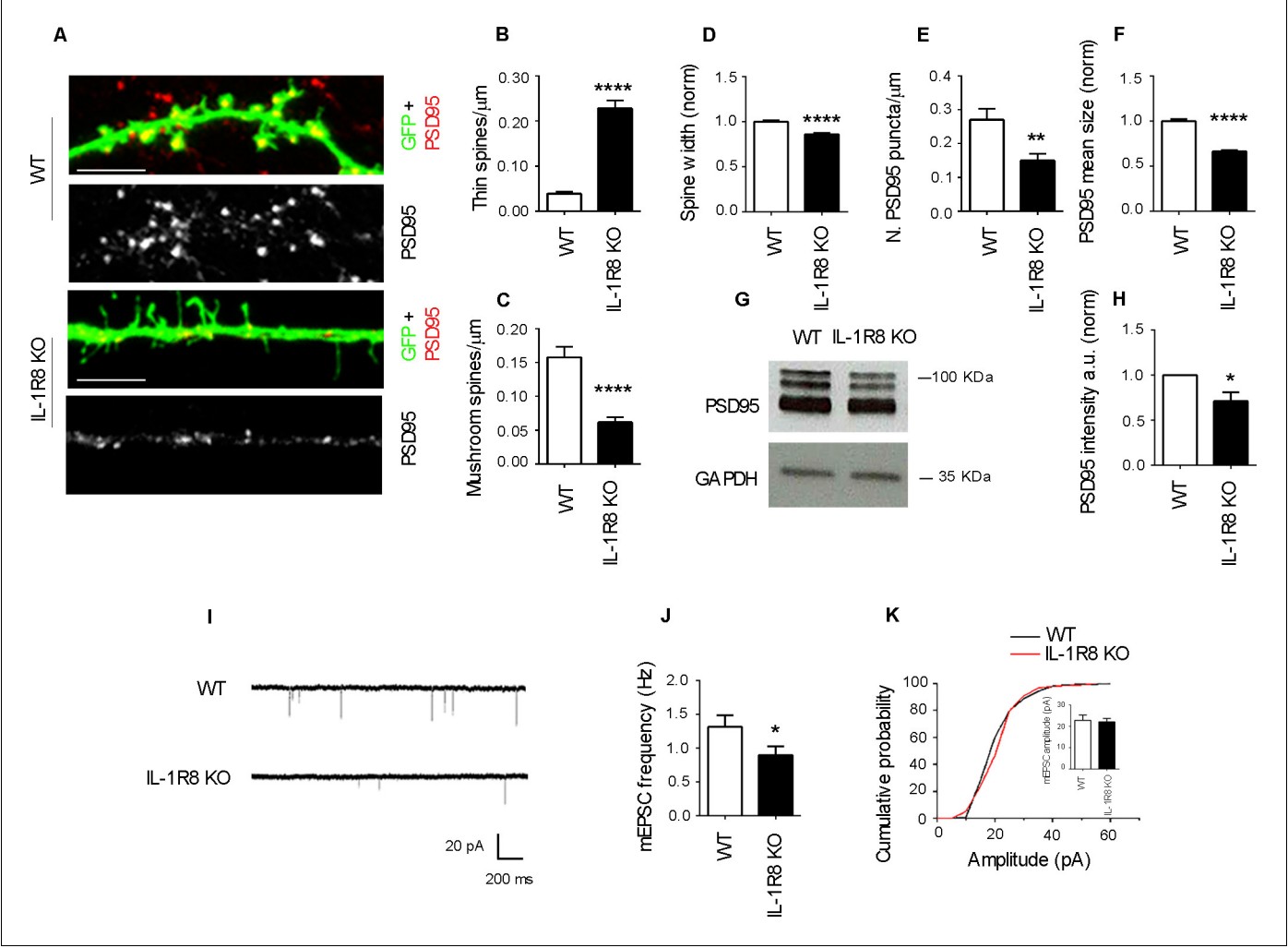

**Figure 1.** IL-1R8 silencing affects spine morphology and function. (A) PSD-95 immunocytochemical staining of GFP-transfected, 16 DIV hippocampal cultures from WT or IL-1R8 KO mice. Scale bar 5 mm. (B-F) Quantitative analysis of the following parameters: thin and mushroom spine density (B and C); spine width (D); PSD-95 puncta density (E) and mean size of PSD-95 puncta (F). Number of analyzed neurons: **B-D**: 32 (WT), 44 (IL-1R8 KO); **E-F**: 32 (WT), 29 (IL-1R8 KO); Student t test. (G, H) Western blotting analysis of PSD-95 levels in primary hippocampal cultures, 3 independent experiments, Mann Whitney test. (I) Representative mEPSC traces recorded from WT and IL-1R8 KO neurons. (J) mEPSC frequency quantitation (WT: n = 12; IL-1R8 KO: n = 18; Mann Whitney Test). (K) Cumulative distributions and bar graph of mEPSC amplitude (WT: 22,82 ± 2, n = 12; IL-1R8 KO: 22,11 ± 1,5, n = 18; Mann Whitney test. * indicates significance compared to WT, # indicates, significance compared to IL-1R8 KO.

application of the c-LTP protocol. Therefore, the IL-1R8 KO synaptic phenotype results from hyper-activation of the IL-1R pathway as a result of IL-1R8 silencing. Consistently, pharmacological activation of IL-1R pathway in WT neurons through overnight treatment with IL-1β (40 ng/ml for 14 hr) resulted in an increased number of immature thin spines and a decreased number of mature mushroom-type spines (*Figure 4G–I*) and PSD-95 puncta (*Figure 4G and J*), accompanied by inability to undergo LTP (*Figure 4G–J*).

Of note, exposure of WT neurons to IL-1Ra prevented LTP, as assessed by confocal analysis (*Figure 4A–C*) or electrophysiological recording (*Figure 4D–F*), indicating that IL-1R acts positively in supporting LTP, even when IL-1R8 expression is not perturbed. In line with these observations, we found that neurons genetically devoid of IL-1 receptor (IL-1R KO) were unable to undergo plasticity phenomena (mushroom spine density: WT, no LTP: 0,13 ± 0009, n = 19; WT, + LTP: 0,36 ± 0,03, n = 18; Student t test, p<00001. IL-1R KO, no LTP: 0,41 ± 0,03; IL-1R KO, + LTP: 0,49 ± 0,03 n = 18; Student t test, p ns = 0,1024). Therefore, in line with literature data (*Costello et al., 2011*;

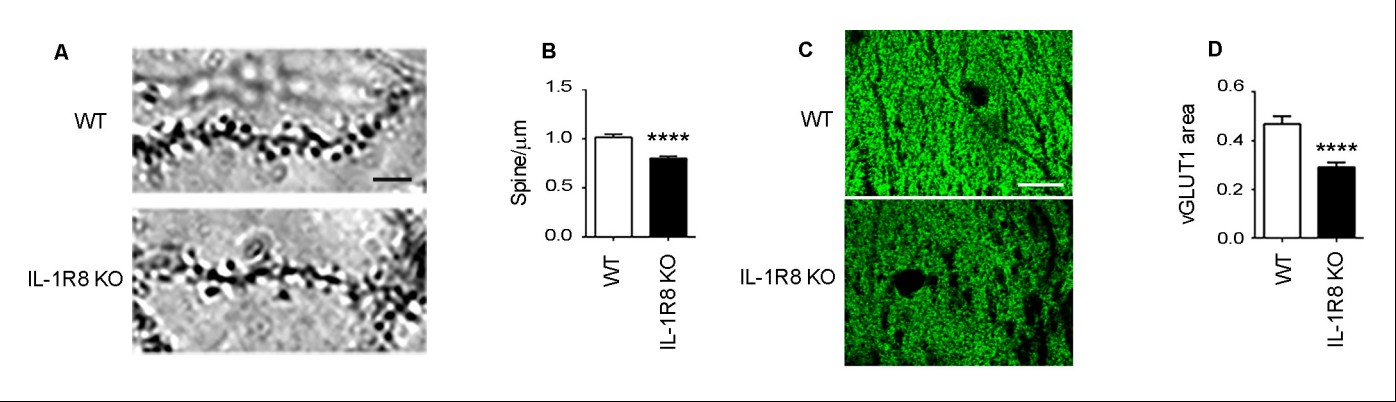

**Figure 2.** IL1R8 deficient mice show altered spines and synapses in hippocampal sections. (**A**) Representative images of secondary branches of apical dendrites of WT and IL-1R8 KO mice (3 months old) stained by the Golgi-Cox method and relative quantitation (**B**). A significant reduction of spine density in IL-1R8 KO mice was evident with respect to WT mice (number of spines per micron: WT = 1,02 ± 0,03; number of mice analyzed: 3, number of examined dendrites: 75; IL-1R8 KO = 0,80 ± 0,02; number of mice analyzed: 3, number of examined dendrites: 84; Mann-Whitney test). Scale bar, 5 µm. (**C**) Representative fields of the CA1 hippocampal region (stratum radiatum) of a WT and IL-1R8 KO mouse brain (1 month old mice), stained for the vesicular glutamate transporter, vGLUT1. Scale bar, 15 µm. (**D**) A significant reduction in vGLUT1 area was found in the stratum radiatum of CA1 field of IL-1R8 KO mice (total area of vGLUT1 positive puncta WT = 0,4644 ± 0,03420; number of examined fields: 35; IL-1R8 KO = 0,2910 ± 0,01984; number of examined fields: 47; Mann-Whitney test).

*Schneider et al., 1998*; *Coogan and O'Connor, 1999*; *Avital et al., 2003*), either pharmacological or genetic silencing of IL-1R is per se sufficient to alter dendritic spine morphology and plasticity, indicating that physiological levels of IL-1R activation are required for correct long-term potentiation.

## IL-1β treatment and IL-1R8 deficiency trigger overlapping gene programs related to hippocampal development and synaptic transmission

To further dissect changes in WT, IL-1R8 KO and WT mice treated with IL-1β, we conducted transcriptomic analysis on cortical tissues. RNA-seq data revealed that genetic ablation of IL-1R8 and pharmacological activation of IL-1R both lead to altered transcription of a shared subset of genes (*Figure 5A and B*). Treatment of WT mice with IL-1β led to transcriptional alterations in 1084 genes (60.6% downregulated and 39.4% upregulated, *Supplementary file 1*). IL-1R8 KO mice showed alterations in expression of 639 genes compared to WT (49.1% downregulated and 50.9% upregulated, *Supplementary file 2*). Comparing the two sets of genes, we observed a highly significant overlap of 193 genes whose expression was altered by either IL-1β treatment and IL-1R8 deficiency (Fischer's exact test; P value = 2.2e-16; Pearson correlation coefficient = 0.95; 61.7% downregulated and 38.3% upregulated, *Supplementary file 3*). Gene Ontology (GO) enrichment analysis revealed that the deregulated genes are clustered in specific categories referring to biological processes such as *hippocampal development* and *synaptic transmission*, among others (*Supplementary file 4*). Moreover, consistent with our results in neuronal cultures, transcriptomic analysis of IL-1R8 KO mice treated with Anakinra revealed that this pharmacological treatment largely reversed the transcriptional alterations observed in IL-1R8 KO mice (83.1%; False Discovery Rate/FDR <0.1; 80.8%, FDR < 0.05), including most of the shared changed genes between IL-1R8 KO mice and in IL1β stimulated mice (89.0%; FDR < 0.1; 88.7%, FDR < 0.05) (*Figure 5C and D*, *Figure 5—figure supplement 1* and *Supplementary file 3*). Notably, the reversion affected important synaptic genes downregulated in both IL-1R8 KO and IL-1β-treated mice, such as *Mdga1* (*Figure 5—figure supplement 1*), *Cnih2* and *Sez6* (*Supplementary file 3*). These data indicate that acute IL-1β treatment and IL-1R8 deficiency trigger overlapping gene programs which are reversed, in the case of IL-1R8 KO mice, by acute exposure to Anakinra.

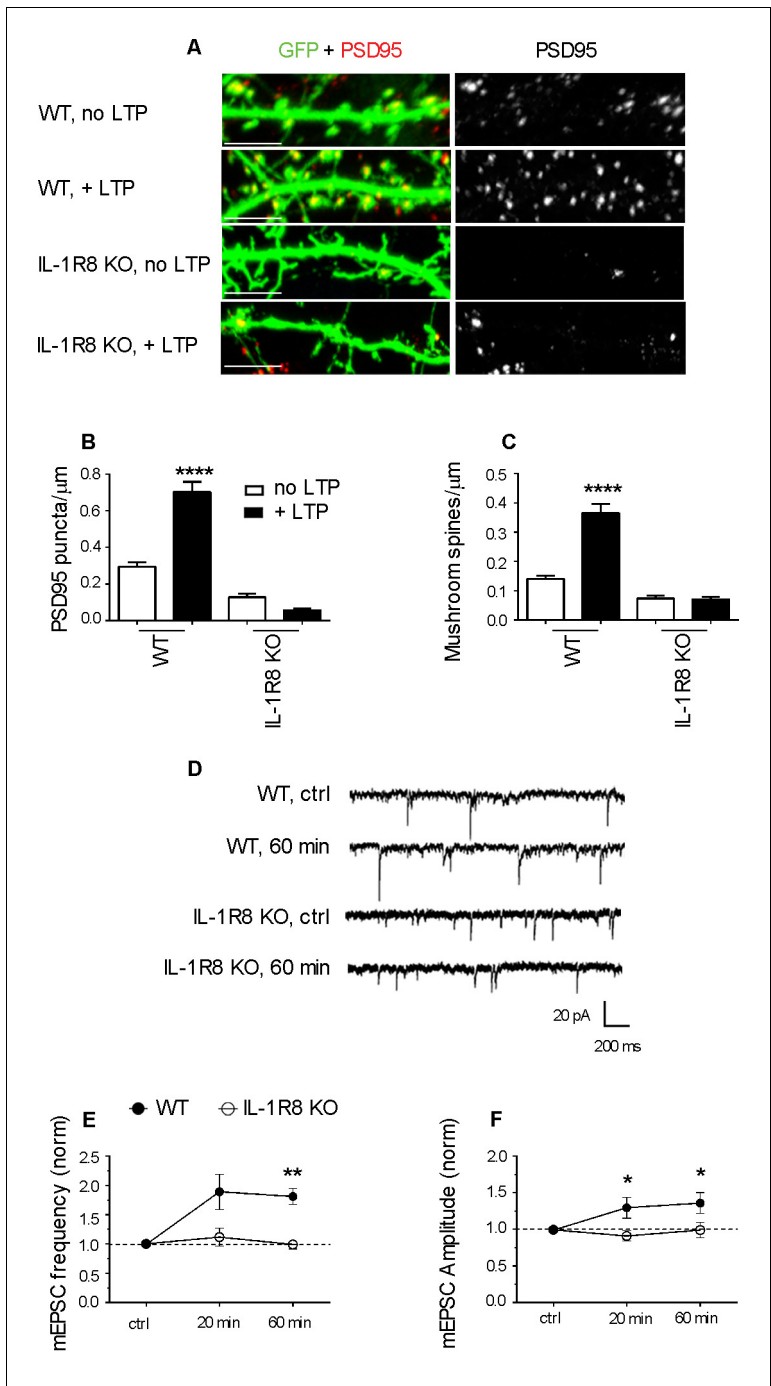

**Figure 3.** IL-1R8 KO neurons do not undergo LTP. (**A**) PSD-95 immunocytochemical staining of GFP-transfected, DIV 16 hippocampal cultures from WT or IL-1R8 KO mice, subjected or not to the LTP protocol. Scale bar 5 μm. (**B** and **C**) Quantitative analysis of PSD-95 and mushroom spine density of neurons treated as above. Number of analyzed neurons, B: 15 (WT, no LTP), 13 (WT, + LTP), 28 (IL-1R8 KO, no LTP), 18 (IL-1R8 KO, + LTP); C: 16 (WT, no LTP), 33 (WT, + LTP), 24 (IL-1R8 KO, no LTP), 34 (IL-1R8 KO, + LTP); one-way ANOVA analysis of variance followed by post hoc Tukey test). (**D**) Representative traces of mEPSCs recorded from neurons of WT or IL-1R8 KO mice before and 60 min after LTP induction. (**E** and **F**) Averaged mEPSC frequency and amplitude of WT and IL-1R8 KO neurons over different recording time points after LTP induction. Normalized mEPSC frequency: WT 0 min: 0,99 ± 0,06, n = 11; 20 min 1,84 ± 0,3, n = 12; 60 min 1,81 ± 0,14, n = 9; IL-1R8 KO 0 min 1,0 ± 0,07, n = 10; 20 min 1,11 ± 0,15, n = 11; 60 min 0,94 ± 0,07, n = 5. Normalized mEPSC amplitude: WT 0 min: 0,99 ± 0,03, n = 11; 20 min 1,29 ± 0,14, n = 12; 60 min 1,36 ± 0,14, n = 9; IL-1R8 KO 0 min 0,96 ± 0,05, n = 10; 20 min 0,91 ± 0,07, n = 11; 60 min 0,99 ± 0,1, n = 5. Mann Whitney test.

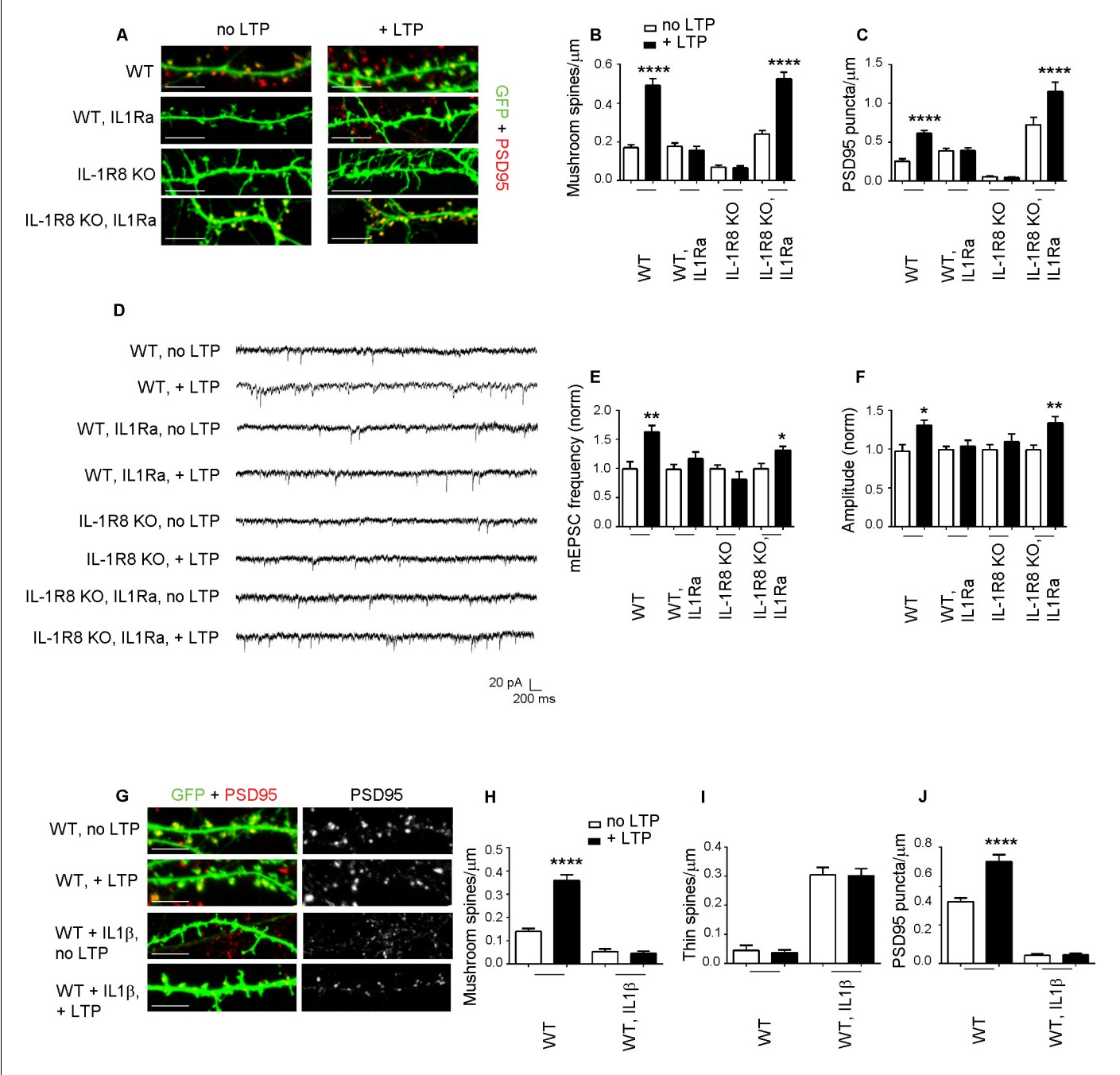

**Figure 4.** Inhibition of IL-1R signalling restores LTP in IL-1R8 KO neurons. (**A**) PSD-95 immunocytochemical staining of 16 DIV hippocampal cultures from GFP transfected WT or IL-1R8 KO mice, treated or not, at DIV 15 with IL1Ra (20 ng/ml) overnight (14 hr). Scale bar 5 μm. (**B**) Quantitative analysis of mushroom spine density in control or upon LTP protocol application. Similar results were obtained with IL1Ra at 100 ng/ml. Number of analyzed neurons: 14 (WT, no LTP), 16 (WT, + LTP), 10 (WT, IL1Ra, no LTP), 10 (WT, IL1Ra, + LTP), 26 (IL-1R8 KO, no LTP), 29 (IL-1R8 KO, + LTP), 23 (IL-1R8 KO, IL1Ra, no LTP), 32 (IL-1R8 KO, IL1Ra, + LTP); one-way ANOVA analysis of variance followed by post hoc Tukey test). (**C**) Quantitative analysis of PSD-95 immunoreactivity. Number of analyzed neurons: 9 (WT, no LTP), 26 (WT, + LTP), 27 (WT, IL1Ra, no LTP), 24 (WT, IL1Ra, + LTP), 24 (IL-1R8 KO, no LTP), 62 (IL-1R8 KO, + LTP), 13 (IL-1R8 KO, IL1Ra, no LTP), 12 (IL-1R8 KO, IL1Ra, + LTP). One-way ANOVA analysis of variance followed by post hoc Tukey test. (**D**) Representative traces of WT and IL-1R8 KO neurons treated with vehicle or IL1Ra (100 ng/ml). (**E** and **F**) Quantitation of mEPSC frequency and amplitude recorded 60 min after LTP protocol in WT or IL-1R8 KO neurons, treated as above. Analysis of normalized mEPSC frequency and amplitude reveals that only WT neurons and IL1-Ra-treated IL-1R8 KO neurons undergo LTP. Number of recorded neurons: 6 (WT, no LTP), 6 (WT, + LTP), 8 (WT, IL1Ra, no LTP), 8 (WT, IL1Ra, + LTP), 9 (IL-1R8 KO, no LTP), 6 (IL-1R8 KO, + LTP), 8 (IL-1R8 KO, IL1Ra, no LTP), 12 (IL-1R8 KO, IL1Ra, + LTP), Mann-Whitney test. (**G**) Immunocytochemical staining for PSD-95 in GFP-transfected WT neurons treated or not with IL-1β (40 ng/ml, overnight) and subjected

*Figure 4 continued on next page*

*Figure 4 continued*
or not to LTP stimulation. Scale bar 5 μm. (**H** and **I**) Quantitative analysis of mushroom and thin spine density. Number of analyzed neurons: 14 (WT, no LTP), 16 (WT, + LTP), 28 (WT, IL-1β, no LTP), 27 (WT, IL-1β, + LTP), one-way ANOVA analysis of variance followed by post hoc Tukey test. (**J**) Quantitative analysis of PSD-95 density. Number of analyzed neurons: 9 (WT, no LTP), 13 (WT, + LTP), 15 (WT, IL-1β, no LTP), 15 (WT, IL-1β, + LTP), one-way ANOVA analysis of variance followed by post hoc Tukey test. Data indicate that application of IL-1β prevents synaptic potentiation.

## IL-1R effects on hippocampal synapses are mediated by the PI3K/AKT/mTOR pathway

We next identified the downstream signaling pathway through which hyperactivation of the IL-1R pathway in IL-1R8 KO neurons influences spine morphology and function. The signaling pathways downstream of IL-1R8 have been described in non-neuronal cells and found to include i) the inhibition of NF-κB and JNK activation, dependent on IL-Rs or TLRs family member activation and ii) the IL-1-dependent activation of the Akt-mTOR pathway (reviewed in *Riva et al. [2012]*). Based on the known involvement of the phosphatidylinositol 3-kinase (PI3K)/Akt/mTOR pathway in the induction and maintenance of LTP in different brain regions (*Opazo et al., 2003*; *Lee et al., 2011*), we focused on the possibility that the lack of IL-1R8 could result in the hyperactivation of this pathway. This possibility would be in line with the demonstrated cognitive and plasticity deficits occurring in genetic mouse models characterized by increased mTOR signaling (reviewed in *Hoeffer and Klann [2010]*). Overnight treatment of 15 DIV IL-1R8 KO neurons with 20 nM rapamycin (*Vézina et al., 1975*), which inhibits both mTORC1 and mTORC2, was sufficient to reverse the defective spine phenotype, with IL-1R8 KO treated neurons displaying an increase in density of both mushroom spines (*Figure 6A and B*) and PSD-95 puncta (*Figure 6A and C*) relative to untreated IL-1R8 KO neurons. Similarly, overnight exposure to 30 μM LY294002, a specific inhibitor of PI3K, or 20 nM wortmannin, a PI3-K/Akt signal transduction inhibitor, rescued mushroom spine and PSD-95 densities in IL-1R8 KO neurons (*Figure 6*). None of the blockers produced damage to the neurons, at least for the time of incubation used. These data demonstrate that hyperactivation of IL-1R in IL-1R8 KO neurons impairs spine morphogenesis and plasticity through the PI3K/AKT/mTOR pathway. Again, WT neurons exposed to the different inhibitors displayed an increase in spine and postsynaptic marker density (*Figure 6A–C*). These data indicate that the endogenous activation of mTOR/PI3K/Akt pathway, like already shown for IL-1R activation (*Figure 4A–C*), is required for correct spine morphogenesis.

## The transcriptional regulator MeCP2 mediates the alterations in spine morphogenesis, synaptic transmission and synaptic plasticity observed in IL-1R8 KO hippocampal neurons

It is known that mTOR is at the cross road of plasticity, memory and disease processes (*Hoeffer and Klann, 2010*). Reduced AKT/mTOR signaling and protein synthesis dysregulation has been described in the brain of the MeCP2 KO Rett syndrome animal model, thus indicating that MeCP2 is an upstream regulator of the Akt/mTOR pathway (*Ricciardi et al., 2011*). Western blotting and immunofluorescence analysis of MeCP2 in IL-1R8 KO hippocampal cultures revealed higher levels of the protein with respect to controls (*Figure 7A and B*), in the absence of differences in the transcript levels (normalized counts WT = 4857 ± 97.46, WT+IL1β = 4936 ± 281.3, IL-1R8 KO = 4864 ± 201.7, IL-1R8 KO+Anakinra = 3599 ± 232.6, mean ± SEM, p>0,05, one-way ANOVA, Tukey's multiple comparisons test). However, relative to previous reports (*Ricciardi et al., 2011*), we found that IL-1R activation and mTOR pathway can be also an upstream regulator of MeCP2. Indeed, overnight treatment of IL-1R8 KO neurons with either IL1Ra (20 or 100 ng/ml) or rapamycin (20 nM) resulted in the normalization of MeCP2 levels, as assessed by Western Blotting (*Figure 7A and B*) or immunofluorescence (*Figure 7C and D*), thus indicating that MeCP2 is increased in IL-1R8 KO neurons as a consequence of the hyperactivation of IL-1R and downstream to the mTOR pathway. Finally MeCP2 levels were increased by immunofluorescence (*Figure 7E and F*) and by western blotting (*Figure 7G and H*) also in cultured WT neurons exposed to IL-1β (40 ng/ml for 14 hr). As a further support to this view, a reduction of MeCP2 intensity was detected in WT neurons exposed to either IL-1Ra or rapamycin and examined by immunofluorescence (*Figure 7C and D*). Consistently, MeCP2 levels, analyzed by Western Blotting, were significantly lower in both hippocampus and cortex of adult IL-

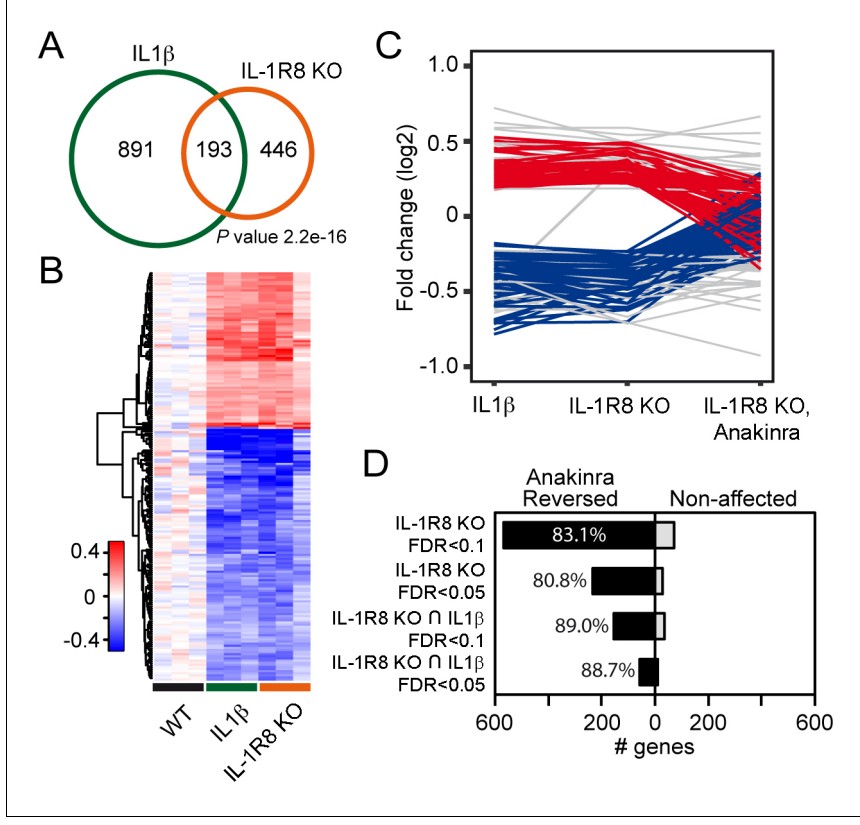

**Figure 5.** Transcriptomic analysis of cortices from WT mice treated with IL-1β and IL-1R8 KO mice reveals common genes with altered regulation, and reversal of altered expression upon treatment of IL-1R8 KO mice with the IL-1β antagonist Anakinra. (**A**) Venn diagram showing significant overlap in the number of Differentially Expressed (DE) genes between conditions (P value = 2.2e-16, Fischer's exact test). IL1β labels DE genes after IL1β administration and IL-1R8 KO show DE genes in IL-1R8 KO mice. (**B**) Heatmap showing a hierarchical clustering of the genes (rows) based on fold changes of expression in each sample versus the average level in the WT condition. Color sidebar for the samples is indicative of the condition: WT (black), treatment with IL-1β (green), or IL-1R8 KO (orange). The inset key shows the color scale of the fold change matrix (log2 values), from blue (downregulated genes) to red (upregulated genes), and white for non-regulated genes. (**C**) Line chart showing fold change (log2 values) against condition (IL1β, IL-1R8 KO, IL-1R8 KO + Anakinra) for DE genes in IL-1R8 KO mice (FDR < 0.1) and also upon IL1β administration to WT mice (FDR < 0.1, 193 genes). Genes that were reversed upon treatment of IL-1R8 KO mice with Anakinra are shown in red (96 upregulated genes) or blue (60 downregulated genes). Genes that were not reversed by Anakinra are shown in light grey (37 genes). (**D**) Bar graph showing the absolute number of genes that were either reversed or non-affected by treatment of IL-1R8 KO mice with Anakinra. Percentage of reversed genes over the total of DE genes in each gene list is also shown. DE gene lists are as follows: IL-1R8 KO FDR < 0.1 (639 genes); IL-1R8 KO FDR < 0.05 (264 genes); IL-1R8 KO ∩ IL1β FDR < 0.1 (193 genes); IL-1R8 KO ∩ IL1β FDR < 0.05 (71 genes). Reversed genes are defined as not found to be DE using the indicated significance threshold in each group, in the comparison of IL-1R8 KO + Anakinra versus WT mice, and those that were differentially expressed in this comparison but in the opposite direction found in IL1β and IL-1R8 KO conditions.

The following figure supplement is available for figure 5:

**Figure supplement 1.** RNA-seq profiles in the cortex of WT, WT treated with IL-1β, IL-1R8 KO mice, and IL-1R8 KO mice treated with Anakinra.

1R KO mice (7 months old) (normalized integrated density of MeCP2 levels in hippocampus, WT: 1 ± 0,0535 (n = 5); IL-1R KO: 0795 ± 0,0121 (n = 5), Student t test, p<0,01. Normalized integrated density of MeCP2 levels in cortex, WT: 1 ± 0,0586 (n = 5); IL-1R KO: 0,7792 ± 0,0473 (n = 5), Student t test, p<0005).

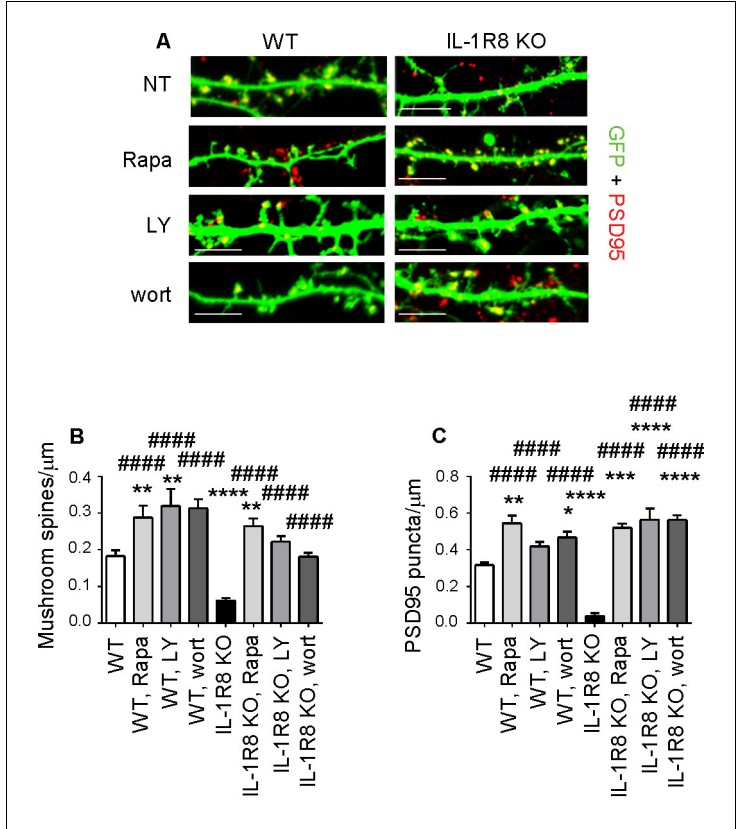

**Figure 6.** Inhibition of mTOR signalling restores LTP in IL-1R8 KO neurons. (**A**) PSD-95 immunocytochemical staining of GFP-transfected 16 DIV hippocampal cultures from WT or IL-1R8 KO mice. At DIV 15 neurons were treated with rapamycin (Rapa, 20 nM), LY294002 (LY, 30 μM) or wortmannin (wort, 20 nM) overnight (14 hr). Scale bar, 5 μm. (**B**) Quantitative analysis of mushroom spine density. Number of analyzed neurons: 36 (WT), 17 (WT, Rapa), 10 (WT, LY), 5 (WT, Wort), 42 (IL-1R8 KO), 35 (IL-1R8 KO, Rapa), 28 (IL-1R8 KO, LY), 30 (IL-1R8 KO, Wort); one-way ANOVA analysis of variance followed by post hoc Tukey test. (**C**) Quantitative analysis of PSD-95 puncta density. Number of analyzed neurons: 17 (WT), 7 (WT, Rapa), 9 (WT, LY), 11 (WT, Wort), 9 (IL-1R8 KO), 15 (IL-1R8 KO, Rapa), 15 (IL-1R8 KO, LY), 18 (IL-1R8 KO, Wort); one-way ANOVA analysis of variance followed by post hoc Tukey test. Data indicate that inhibitors of the mTOR pathway restore synaptic potentiation. * indicates significance compared to WT, # indicates significance compared to IL-1R8 KO.

MeCP2, a synaptic factor that controls spine morphogenesis and plasticity (*Chao et al., 2007*; *Nelson et al., 2006*; *Zoghbi, 2003*; *Asaka et al., 2006*; *Guy et al., 2007*; *Moretti et al., 2006*), needs to be tightly regulated in the human brain. Indeed, besides the well-known pathological traits - intellectual disability and delayed development - caused by MeCP2 duplication (*Ramocki et al., 2010*) or MeCP2 loss of function (*Chahrour and Zoghbi, 2007*), even mild differences in MeCP2 expression turned out to profoundly impact human behavior and brain function (*Tantra et al., 2014*). To investigate whether the morphological and functional defects observed in the IL-1R8 KO neurons could directly result from increased MeCP2 levels, we silenced MeCP2 in neuronal cultures from IL-1R8 KO mice through the use of a well characterized shRNA construct (Sh MeCP, kind gift of Dr. M. Greenberg, Harvard Medical School), which has been widely used in literature to reduce MeCP2 expression (*Zhou et al., 2006*; *Blackman et al., 2012*; *Gangisetty et al., 2014*; *Kishi et al., 2016*; *Bedogni et al., 2016*). DIV 12 neurons were transfected with shCTRL or shMeCP2, and examined for their ability to undergo LTP four days later. We first quantitated the extent of MeCP2 reduction in treated cultures and found that the use of shRNA construct allowed to achieve a protein reduction ranging from 59% to 67%. However, given the higher MeCP2 expression in IL-1R8 KO neurons, the levels of MeCP2 in shRNA–treated IL-1R8 KO neurons ended up to be comparable to those in untreated WT neurons (*Figure 8A and B*). While WT neurons with silenced MeCP2 showed

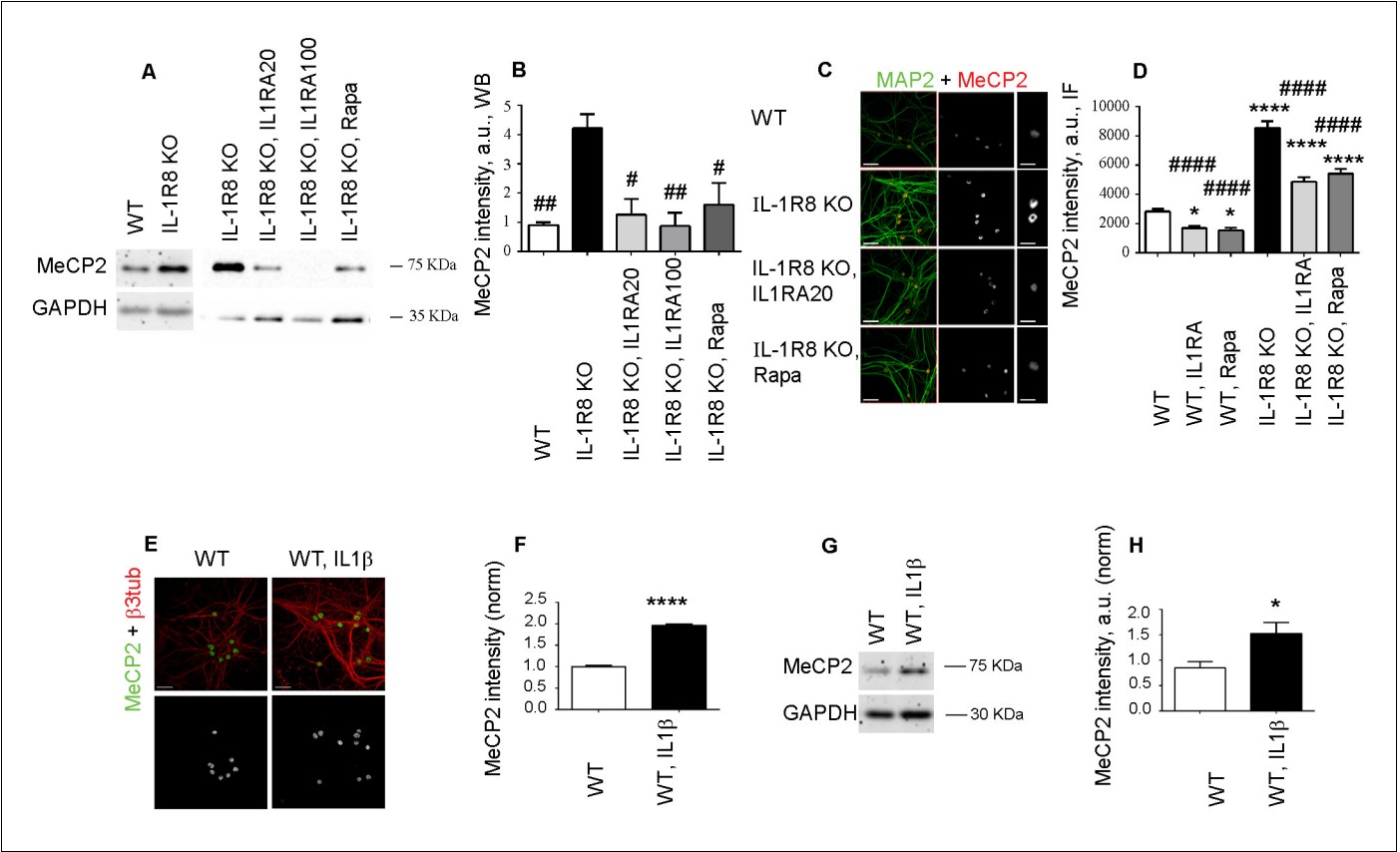

**Figure 7.** IL-1R8 KO neurons display higher MeCP2 levels that are responsible for LTP defects in IL-1R8 KO neurons. (A) MeCP2 expression analyzed by western blotting in cultured neurons from WT or IL-1R8 KO neurons treated or not with IL-1Ra (20 ng/ml), IL-1Ra (100 ng/ml) or Rapa (20 nM). A higher MeCP2 expression is detectable in IL-1R8 KO neurons, which is reduced by IL-1Ra or Rapa. (B) Quantitative analysis of MeCP2 expression. Number of replicates: 3 (WT), 3 (IL-1R8 KO), 3 (IL-1R8 KO, IL1Ra 20), 3 (IL-1R8 KO, IL1Ra 100), 3 (IL-1R8 KO, Rapa); one-way ANOVA analysis of variance followed by post hoc Tukey test. (C) MeCP2 and MAP2 immunocytochemical staining of 16 DIV hippocampal WT or IL-1R8 KO neurons treated overnight (14 hr) with vehicle, IL1Ra (20 ng/ml) or Rapa (20 nM). Scale bar, 40 μm (low magnification image), 20 μm (insert). (D) Quantitative analysis of MeCP2 immunoreactivity in neurons treated as above. Number of analyzed neurons: 88 (WT), 60 (WT, IL1Ra), 55 (WT, Rapa), 52 (IL-1R8 KO), 66 (IL-1R8 KO, IL1Ra), 81 (IL-1R8 KO, Rapa); one-way ANOVA analysis of variance followed by post hoc Tukey test. (E) Immunocytochemical staining for MeCP2 and $\beta$3-tubulin of 16 DIV WT hippocampal neurons, exposed or not to IL1$\beta$ (40 ng/ml) overnight. Scale bar, 40 μm. (F) Quantitative analysis of MeCP2 immunoreactivity reveals higher MeCP2 levels in IL1$\beta$ treated neurons. Number of analyzed neurons: 288 (WT), 227 (WT, IL1$\beta$); Student t test. (G and H) Representative western blotting (G) and quantitative analysis (H) of MeCP2 expression in WT neurons treated with IL-1$\beta$ (40 ng/ml). Number of replicates, WT: n = 5; WT + IL1$\beta$ 40: n = 5 independent experiments. Mann Whitney test. * indicates significance compared to WT, # indicates significance compared to IL-1R8 KO.

abnormal spine morphology, with a decreased density of mushroom spines (*Figure 8C and D*) and PSD95 puncta (*Figure 8C and E*), IL-1R8 KO neurons with silenced MeCP2 reverted the morphological phenotype. Even more interestingly, MeCP2 silenced neurons rescued their ability to undergo LTP (*Figure 8F–H*). Therefore, recovery of MeCP2 to control levels is sufficient to acutely rescue the synaptic defects observed in IL-1R8 KO neurons.

## Increased MeCP2 levels are detectable in the cortex and hippocampus of IL-1R8 deficient mice

The increased MeCP2 expression detected in primary IL-1R8 KO neurons was also confirmed in vivo. Immunohistochemical analysis of hippocampus (CA1 region) and cortex revealed a significantly higher expression of MeCP2 in IL-1R8 KO neurons compared to WT, as indicated by the quantitation of the MeCP2 integrated density value per neuron and the cumulative distribution of neuronal integrated density values (*Figure 9A–E*). Consistently, western blot analysis of either hippocampus or

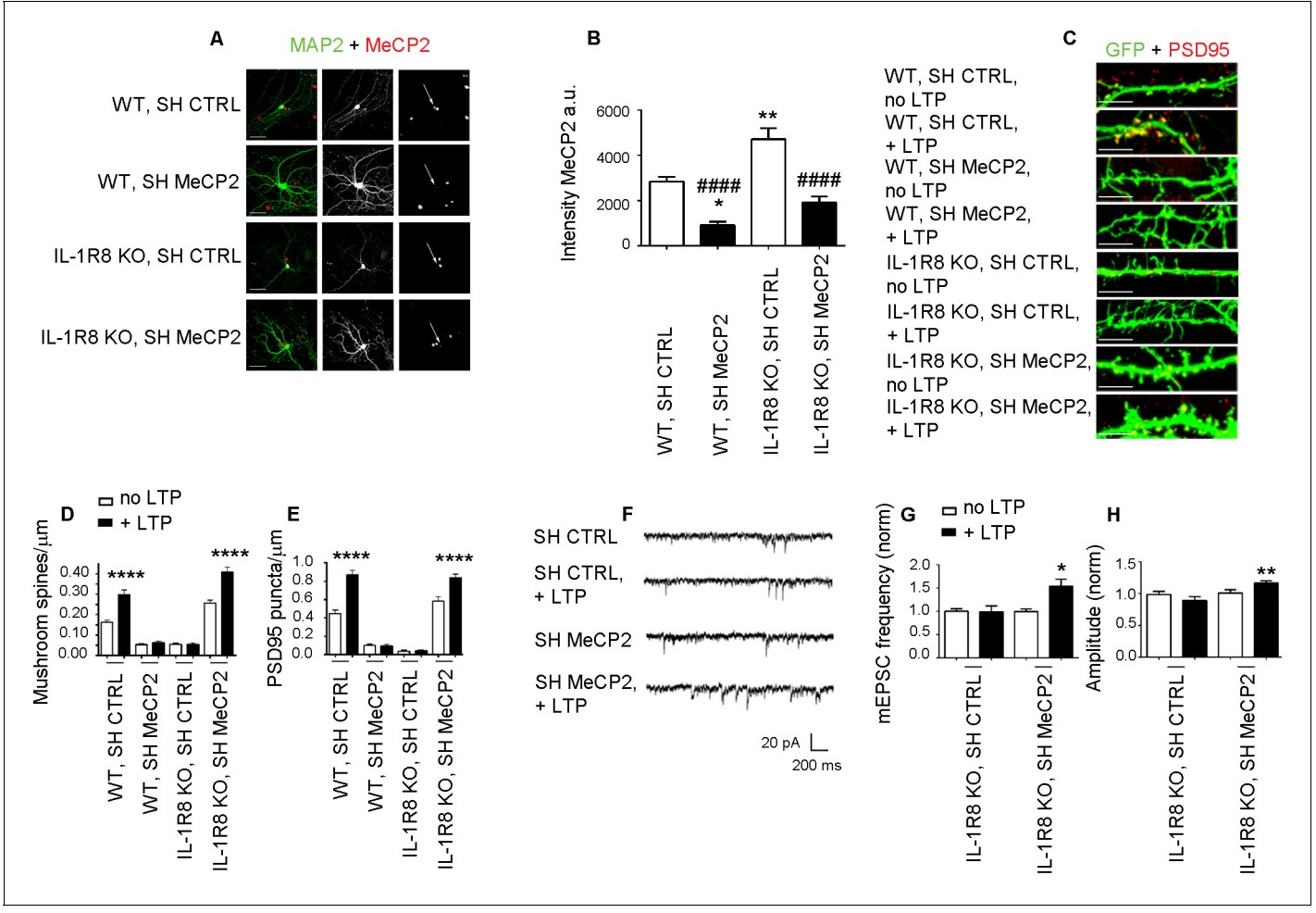

**Figure 8.** Higher MeCP2 levels are responsible for LTP defects in IL-1R8 KO neurons. (**A**) MeCP2 and MAP2 immunocytochemical staining and (**B**) quantitative analysis of MeCP2 immunoreactivity in 16 DIV WT and IL-1R8 KO neurons transfected with SH CTRL or SH MeCP2. Scale bar 40 µm. Number of analyzed neurons: 23 (WT, SH CTRL), 28 (WT, SH MeCP2), 46 (IL-1R8 KO, SH CTRL), 40 (IL-1R8 KO, SH MeCP2); one-way ANOVA analysis of variance followed by post hoc Tukey test. * indicates significance compared to WT + SH CTRL, # indicates significance compared to IL-1R8 KO + SH CTRL. (**C**) PSD-95 immunocytochemical staining of SH CTRL- or SH MeCP2-transfected, DIV 16 hippocampal cultures from WT or IL-1R8 KO mice. Neurons were transfected at DIV 12 and the LTP protocol was applied at DIV 16. Scale bar, 5 µm. (**D**) Quantitative analysis of mushroom spine density in WT and IL-1R8 KO neurons treated as above. Number of analyzed neurons: 30 (WT, SH CTRL, no LTP), 29 (WT, SH CTRL, + LTP), 30 (WT, SH MeCP2, no LTP), 29 (WT, SH MeCP2, + LTP), 59 (IL-1R8 KO, SH CTRL, no LPT), 60 (IL-1R8 KO, SH CTRL, + LTP), 54 (IL-1R8 KO, SH MeCP2, no LTP), 58 (IL-1R8 KO, SH MeCP2, + LTP); one-way ANOVA analysis of variance followed by post hoc Tukey test. (**E**) Quantitative analysis of PSD-95 density. Number of analyzed neurons: 17 (WT, SH CTRL, no LTP), 15 (WT, SH CTRL, + LTP), 16 (WT, SH MeCP2, no LTP), 15 (WT, SH MeCP2, + LTP), 15 (IL-1R8 KO, SH CTRL, no LTP), 15 (IL-1R8 KO, SH CTRL, + LTP), 15 (IL-1R8 KO, SH MeCP2, no LTP), 17 (IL-1R8 KO, SH MeCP2, + LTP); one-way ANOVA analysis of variance followed by post hoc Tukey test. Data indicate that reduction of MeCP2 expression restores synaptic potentiation. (**F**) Representative traces of mEPSC recorded from IL-1R8 KO neurons transfected with SH CTRL or SHMeCP2 before and after LTP induction. (**G** and **H**) Quantitation of mEPSC frequency and amplitude of neurons treated as above. Analysis of normalized mEPSC frequency and amplitude reveals that only neurons transfected with SH MeCP2 undergo LTP. Number of recorded neurons: 14 (IL-1R8 KO, SH CTRL, no LTP), 7 (IL-1R8 KO, SH CTRL, + LTP), 19 (IL-1R8 KO, SH MeCP2, no LTP), 7 (IL-1R8 KO, SH MeCP2, + LTP). Mann Whitney test.

cortex revealed higher levels of MeCP2 in the brains of IL-1R8 KO mice (*Figure 9F–H*). Notably, administration of 30 mg/kg Anakinra to IL-1R8 KO mice for three consecutive days significantly reduced MeCP2 protein (*Figure 9F–H*). These data indicate that higher levels of MeCP2 are detectable in the brain of IL-1R8 KO mice and that the acute inhibition of IL-1R by Anakinra reduces hippocampal and cortical MeCP2 levels.

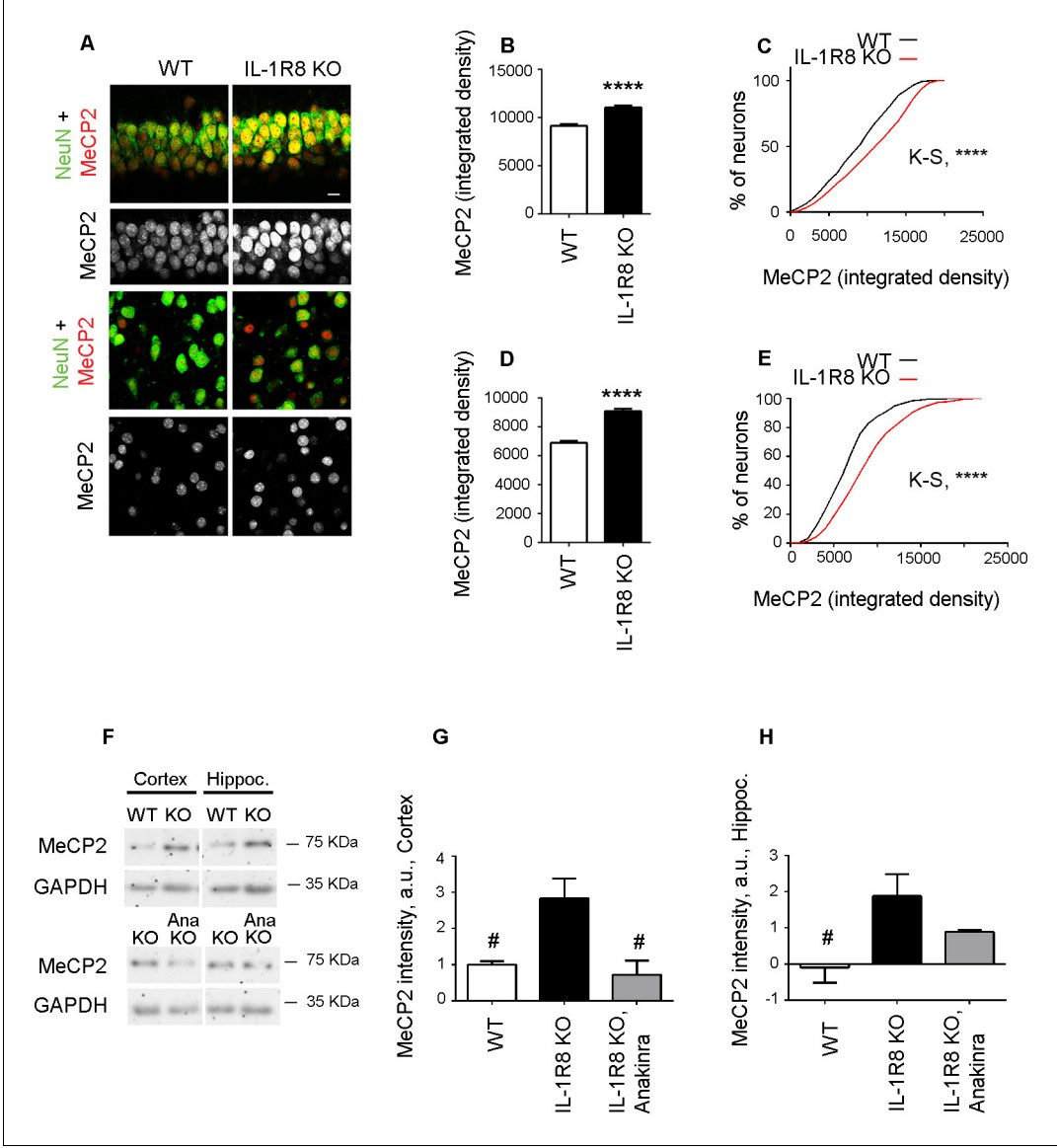

**Figure 9.** Higher MeCP2 levels in the brain of IL-1R8 KO mice. (**A**) Representative images of brain sections (CA1 hippocampus, upper panels and cortex, lower panels) of WT and IL-1R8 KO mice (1 month old) stained for MeCP2 and NeuN, as indicated. (**B-E**) Graphs show quantitation of MeCP2 mean integrated density values and the cumulative distributions of neuronal MeCP2 integrated density values in CA1 hippocampal neurons (**B** and **C**) and cortical neurons (**D** and **E**). Number of analyzed mice and neurons: WT: n = 3 mice, hippocampal neurons = 572, cortical neurons = 481. IL-1R8 KO: n = 3 mice, hippocampal neurons = 559, cortical neurons = 471. Statistical comparison: Mann-Whitney Test for **B** and **D**; KolmogorovSmirnov Comparison (http://www.physics.csbsju.edu/stats/KS-test.html), D values are: 0,2084 with a corresponding p<0.0001 (panel **C**) and 0,1396 with a corresponding p=0.0005 (panel **E**). (**F-H**) Western blotting (**F**) and quantitative analysis (**G** and **H**) of MeCP2 expression in cortices and hippocampi of 3 months old WT and IL-1R8 KO mice or in IL-1R8 KO mice treated with Anakinra (30 mg/kg) for 3 days. Number of analyzed mice: 6 WT, 5 IL-1R8 KO and 5 IL-1R8 KO + Anakinra (**G**); 6 WT, 6 IL-1R8 KO and 4 IL-1R8 KO + Anakinra (**H**). Statistical test: one-way ANOVA analysis of variance followed by post hoc Tukey test.

## Inhibition of IL-1β receptor by Anakinra rescues behavioral defects

We finally investigated whether behavioral deficits consequent to the genetic lack of IL-1R8 could be rescued in the adult mice by the pharmacological inhibition of IL-1R. To this aim IL-1R8 KO mice were analyzed for novel place recognition test three days after the treatment with the anti-

inflammatory compound glycyrrhizic acid (50 mg/kg), which binds to high-mobility group box 1 (HMGB1) protein and inhibits IL-1 activity (*Sakamoto et al., 2001*; *Mollica et al., 2007*) or with 30 mg/kg Anakinra. Both treatments ameliorated the spatial memory impairment of IL-1R8 KO mice (*Figure 10A*), with 50% IL-1R8 KO mice displaying a significantly improved discrimination index after Anakinra treatment (p=0,0168 paired Student's t-test). IL-1R8 KO mice were also tested for the Morris water maze and the rewarded T maze. In line with the concept that alternation, either rewarded or spontaneous, detects hippocampal dysfunction even better than the Morris water maze (reviewed in *Deacon and Rawlins, 2006*), the rewarded T-maze (*Figure 10C*), but not the Morris water maze (*Figure 10B*), revealed an impairment of spatial memory in IL-1R8 KO mice. Treatment with Anakinra ameliorated the performance (*Figure 10C*). Consistent with these data, IL-1R8 KO mice performed a

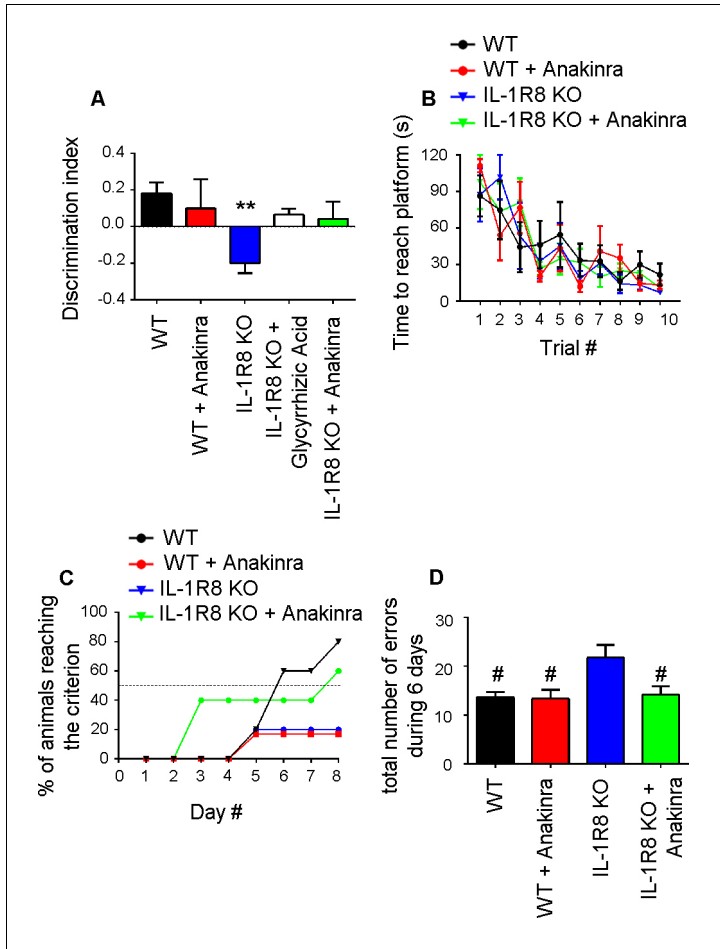

**Figure 10.** Anakinra rescues behavioral defects in IL-1R8 KO mice. (**A**) Analysis of novel-place object recognition task in WT and IL-1R8 KO mice shows a defect in spatial learning in IL-1R8 KO mice (one-way ANOVA followed by Bonferroni multiple comparison test). 3 days i.p. treatment with 50 mg/kg glycyrrhizic acid or 30 mg/kg Anakinra ameliorates IL-1R8 KO mice performance. Number of analyzed mice: 15 (WT), 6 (WT+Anakinra), 18 (IL1-R8 KO), 3 (IL1-R8 KO+glycyrrhizic acid) and 10 (IL-1R8 KO+Anakinra). (**B**) IL-1R8 deficiency or treatment with Anakinra did not affect the learning performance in the Water Maze navigation task. Number of analyzed mice: 5 (WT), 5 (WT +Anakinra), 5 (IL-1R8 KO), and 5 (IL-1R8 KO+Anakinra). (**C**) IL-1R8 KO mice displayed impaired spatial memory in the rewarded T-maze task, as indicated by the low percentage of animals reaching the criterion. Treatment with Anakinra ameliorated the performance of IL1-R8 KO mice, while worsening the learning ability of WT. Number of analyzed mice: 5 (WT), 3 (WT+Anakinra), 6 (IL-1R8 KO) and 5 (IL-1R8 KO+Anakinra). (**D**) IL-1R8 KO mice showed a significant increase in the total number of errors during the acquisition phase as compared to the WT, which was significantly reduced by treatment with Anakinra. One-way ANOVA followed by Tukey's multiple comparison test. Number of analyzed mice: 6 (WT), 5 (WT+Anakinra), 5 (IL1-R8 KO) and 5 (IL-1R8 KO+Anakinra). All the mice were 3–6 months old.

significantly higher number of errors during the acquisition phase with respect to WT and Anakinra-treated IL-1R8 KO mice (*Figure 10D*). Interestingly, also WT mice treated with Anakinra displayed impaired spatial memory (*Figure 10C*). The lack of increased number of errors in WT mice treated with Anakinra (*Figure 10D*) is possibly attributable to the enhancement of freezing behavior. These data indicate that Anakinra ameliorates the spatial memory of IL-1R8 KO mice and confirm that blockade of IL-1R originates learning defects in WT mice.

## Discussion

Finely tuned expression of immune molecules has been found to modulate CNS function (*Deverman and Patterson, 2009*), both throughout normal development and in pathological conditions. During development, immune molecules such as CXCR4, interferon γ, IL-1β, IL-6, IL-9, IL-10 and transforming growth factor β affect neurogenesis, neuronal migration, axon guidance, synapse formation, activity-dependent refinement of circuits and synaptic plasticity (*Deverman and Patterson, 2009*; *Zhao and Schwartz, 1998*). In line with this concept, mice genetically devoid of IL-1R8, a receptor which dampens the activation of the TLRs and IL-1R signalling pathways, are neurologically impaired, showing defects in novel place recognition task, spatial reference memory and LTP, defects that occur in the absence of any external inflammatory stimuli (*Costello et al., 2011*). However, the molecular processes at the basis of these defects are still elusive.

Our study provides two major breakthroughs in the field of the cross talk between nervous and immune systems: first, it demonstrates that hyperactivation of the IL1R pathway results in the overexpression of MeCP2, a synaptic factor that controls spine morphogenesis, synaptic transmission and plasticity, which is responsible for syndromes associated with intellectual disability such as Rett syndrome and MeCP2 duplication syndrome; second it provides the proof-of-concept that the synaptic and behavioral defects consequent to genetic hyperactivation of the IL-1R pathway can be rescued in the adult by pharmacological treatment with IL-1Ra. Of note, the acute application of IL-1β to mature neurons recapitulates similar synaptic defects and alterations of MeCP2 expression. Consistently, transcriptomic analysis of cortices from WT mice (treated or not with IL-1β) and from IL-1R8 KO animals (treated or not treated with Anakinra) show that a common set of genes is transcriptionally altered in WT mice treated with IL-1β and in IL-1R8 KO animals, including genes controlling synaptic function and spine morphogenesis. The expression of this set of genes is restored to normal levels by treatment with Anakinra in IL-1R8 KO mice. These findings have important repercussions in the clinic, as they could open the road to the use of anti-inflammatory drugs as therapeutic treatments for neuropsychiatric illnesses such as schizophrenia, autism and mental retardation, for all of which an immune component has been demonstrated (*Khandaker et al., 2015*; *Theoharides et al., 2015*; *Young et al., 2016*).

Our study identifies the crucial involvement of the AKT/mTOR pathway in IL-1R8 KO neurons (*Figure 10H*). IL-1R8 has been previously reported to regulate mTOR kinase activity in Th17 cells, playing a nonredundant role in controlling mTOR-dependent differentiation, proliferation and cytokine production (*Gulen et al., 2010*). The demonstration that the PI3K/Akt/mTOR pathway is at the root of synaptic defects in IL-1R8 KO neurons is in line with the established involvement of this pathway in dendritic tree development and spine formation in neurons (*Sarbassov et al., 2006*), as well as actin cytoskeleton dynamics and synaptic plasticity (*Jaworski and Sheng, 2006*). Remarkably, the mTOR pathway is closely linked to ILR/TLR signaling since PI3 kinase is required for activation of NF-κB by IL-1 (*Gulen et al., 2010*). Furthermore, the signaling module containing the MyD88 adaptor protein, together with phosphorylated IRAK and TRAF6, which is downstream to both IL-1R and TLRs activation, is essential for PI3K recruitment and Akt activation, making mTOR one of the downstream targets of ILR/TLRs signaling (*Gulen et al., 2010*). Consistent with the involvement of mTOR in the IL-1R8 KO neuron defects, treatment with the PI3 kinase inhibitors, LY294002 and wortmannin restored the neuronal ability to sustain LTP. Moreover, specific inhibition of mTOR by the drug rapamycin rescues synaptic morphology in IL-1R8 KO neurons. These results indicate that the PI3K-regulated pathway is involved in the downstream signaling that connects excessive IL-1R activation to the morphological and functional impairments of dendritic spines observed in IL-1R8 KO neurons.

More importantly, our study shows for the first time that activation of IL-1R in neurons increases the levels of MeCP2, both in vitro and in vivo. Fine-tuning of MeCP2 expression is required for proper synapse function (reviewed in *Na et al. [2013]*). Indeed, lack of expression of MeCP2 results

in deceleration of body and head growth rate, problems in motor and speech capabilities, irregularities in motor activity and difficulties in breathing, and also in cognitive defects characteristic of an autism-spectrum disorder (reviewed in *Percy and Lane [2005]*). Mouse models of MeCP2 duplication display a neurological phenotype, stereotyped and repetitive movements, epilepsy, spasticity, hypoactivity, early death (*Collins et al., 2004*) and defects in dendritic arborization and spine morphology (*Jiang et al., 2013*). In humans, sporadic mutations in the gene coding for MeCP2 results in Rett syndrome (*Amir et al., 1999*), while a double dosage of MeCP2 causes a severe developmental delay and mental retardation (*Lubs et al., 1999*). Thus, MeCP2 levels must be finely tuned as even mild over-expression of this factor can have a robust effect. We first demonstrate here that immune activation controls the expression of MeCP2 in neurons. The possibility that MeCP2 phosphorylation, by which MeCP2 also modulates gene expression levels, is altered in these models is an open issue. MeCP2 can act as a transcriptional activator (when interacting with CREB) but primarily functions as a transcriptional repressor associated with mSin3A and HDACs repressor complexes (*Chahrour et al., 2008*; *Jones et al., 1998*; *Nan et al., 1998*). Consistently, we observed a prevalence of downregulation over upregulation in the expression of the deregulated genes in WT mice treated with IL-1$\beta$ and in IL-1R8 KO animals, suggesting that the repressor function of MeCP2 might dominate in response to IL-1R activity. Indeed, gene ontology enrichment analysis for the deregulated genes highlighted negative regulation of transcription from RNA polymerase II promoter and negative regulation of gene expression among the most affected biological processes in IL-1R8 KO animals (*Supplementary file 4*).

The regulation of MeCP2 protein levels by IL-1R8 appears to result from enhanced activity of IL-1R. IL-1$\beta$ is a fundamental factor of inflammatory responses in the brain. It is expressed at baseline levels in the healthy brain, and its expression increases following peripheral infection, surgery, or brain injury, as well as in neurodegenerative diseases (*Rothwell and Luheshi, 2000*). Increases in IL-1$\beta$ levels lead to cognitive decline, in particular in hippocampal-dependent tasks (*Rachal Pugh et al., 2001*). IL-1$\beta$ overexpression for two weeks in an inducible transgenic mouse was found to cause impairment in long-term contextual and spatial memory, without effects on short-term and non-hippocampal memory (*Hein et al., 2010*). An increase in IL-1$\beta$ levels due to Escherichia coli infection leads to defects in contextual fear conditioning, with loss of memory prevented by IL1Ra (*Barrientos et al., 2009*; *Frank et al., 2010*). Remarkably, we show that neuron treatment with IL-1$\beta$ not only leads to an increase in immature spines unable to undergo LTP, but also augments MeCP2 levels. In this context, as with E. coli infection, IL1Ra/Anakinra restores synaptic function.

Our demonstration that enhanced IL-1R activity, due to either lack of IL-1R8 or excessive IL-1$\beta$ signaling, impacts synapse function through regulating the expression of MeCP2 expands in a critical way previous research on the role of MeCP2 in cognition (*Na et al., 2013*): we show here that MeCP2 is a fundamental node linking inflammation with synaptic damage and we demonstrate that inhibition of IL-1R pathway by IL-1Ra restores synaptic structure and function in vitro. We also demonstrate that the acute treatment with Anakinra, a pharmacological agent currently in use to treat chronic inflammation, restores synaptic plasticity in vivo. Although we do not have the proof that this occurs exclusively through MeCP2 reduction, our data indicate that immune drugs may be efficacious for treating neurological deficits associated with immune pathologies with a genetic basis and justify further research into anti-inflammatory treatment for selected brain pathologies. Indeed, recurrent infections have been found to occur in 70% of individuals affected by MeCP2 duplication syndrome, which lead to further deterioration of the general and neurological status, being even fatal in some patients (*van Esch et al., 2012*). Also, in patients affected by cryopyrin-associated periodic syndrome (CAPS), a group of rare autoinflammatory diseases with genetic basis, the levels of IL-1$\beta$ are fivefold higher than in healthy individuals, leading to persistent unregulated systemic inflammation (*Janssen et al., 2004*; *Lachmann et al., 2009*). CAPS is characterized by recurrent bouts of fever with malaise and chills, urticarial neutrophilic, eye redness due to conjunctivitis, arthralgia and myalgia with intense fatigue. Children affected by CAPS are believed to show symptoms of intellectual disability. Mental and hearing defects are reversed following treatment with Anakinra (*Goldbach-Mansky, 2011*; *Goldbach-Mansky et al., 2006*; *Lepore et al., 2010*; *Miyamae et al., 2010*; *Neven et al., 2010*) and with specific neutralization of IL-1$\beta$ with canakinumab. Our study opens the possibility that further cognitive deterioration may result from the enhanced inflammation and hyperactivation of the IL-1 signaling pathway, which might further increase MeCP2 levels in a harmful positive feedback loop. The challenging possibility that treatment

with Anakinra may help interrupting this spiral and ameliorating the cognitive deficits in affected individuals is worth to be tested.

## Materials and methods

### Animals

IL-1R8 KO mice (*Garlanda et al., 2004*) and double IL-1R8 KO IL-1R KO mice (*Véliz Rodriguez et al., 2012*) were obtained from Istituto Clinico Humanitas IRCCS, Milan, Italy. Primary hippocampal cultures were performed from E17 embryos. Tissues for WB analysis were taken from 3-month-old male animals. All the experimental procedures followed the guidelines established by the European Legislation (Directive 2010/63/EU), and the Italian Legislation (L.D. no 26/2014).

### Golgi staining and quantification of dendritic spines

Mice (3 months old) were deeply anesthetized with chloral hydrate (4%; 1 ml/100 g body weight, i. p.) and perfused intracardially with 0.9% saline solution. The brains were removed and stained by modified Golgi-Cox method as described in (*Menna et al., 2013*) with slight modifications. Coronal sections of 100 μm thickness from the dorsal hippocampus were obtained using a vibratome (VT1000S, Leica, Wetzlar, Germany). These sections were collected free floating in 6% sucrose solution and processed with ammonium hydroxide for 15 min, followed by 15 min in Kodak Film Fixer, and finally were rinsed with distilled water, placed on coverslips, dehydrated and mounted with a xylene-based medium. Spine density was counted on the secondary branches of apical dendrites of pyramidal neurons located in the CA1 subfield of the dorsal hippocampus. At least 30 neurons per animal were evaluated.

### Immunofluorescent staining on free-floating sections

Mice (1 month old) were deeply anesthetized with chloral hydrate (4%; 1 ml/100 g body weight, i.p.) and perfused intracardially with 4% paraformaldehyde. Immunofluorescent staining was carried out on free-floating sections as described in (*Menna et al., 2013*). Free-floating sections at the level of dorsal hippocampus were processed with the specific antibodies as indicated, followed by incubation with the secondary antibodies, counterstained with DAPI and mounted in Fluorsave (Calbiochem, San Diego, CA, USA).

Primary antibodies: anti-vGLUT-1 (guinea pig polyclonal antibody, 1:1000; No. 135 304 Synaptic System), anti-MeCP2 (rabbit polyclonal, 1:200; M9317 Sigma), anti-NeuN (mouse monoclonal, 1:500; MAB377 Millipore). Sections were examined by means of a Zeiss LSM 510 META confocal microscope (Leica Microsystems, Germany). Images were acquired in the stratum pyramidale or stratum radiatum of the CA1 subfield of the hippocampus (as indicated) using the x40 oil immersion lens with an additional electronic zoom factor of up to 3 (voxel sizes of $0.10 \times 0.10 \times 1$ μm) maintaining the parameters of acquisition (laser power, pinhole, gain, offset) constant among groups.

### Behavior

#### Novel place object recognition

This test was used to assess whether mice were able to recognize that an object that they had experienced before had changed location in the arena, as previously described (*Corradini et al., 2012*). The test was conducted in an open plastic arena (60x x50x x30 cm) in two phases. During the first phase, lasting two days, mice were habituated to the test arena for 10 min on the first day. On the second day, mice were subjected to familiarization (T1) and novel place recognition (T2). During familiarization, two identical objects were placed in the North and South corners of the arena. Each animal was placed in the center of the arena, equidistant from the objects, and was left for a maximum of 10 min or until it had completed 30 s of cumulative object exploration, then it was returned to its home cage. 120 min later the mouse was introduced in the same arena with one object relocated to the East instead of the South corner for a maximum of 5 min. Scoring was performed in the same manner as during familiarization.

To assess the role of inflammation in the cognitive performances observed in IL-1R8 KO mice, 30 days later mice were treated with 50 mg/kg i.p. glycyrrhizin acid, 30 mg/kg, i.p. Anakinra or vehicle for 3 days and were re-subjected to the T2.

The objects used consisted of plastic cylinders and colored plastic stacks. The arena and the objects were cleaned with 70% ethanol after each trial. A discrimination index was calculated to evaluate the performance of each mouse as (N-F)/(N+F), where N = time spent exploring the object in the new location during T2, and F = time spent exploring the object in the familiar location during T2. Mice who did not move in the arena were excluded from the analysis.

## Water maze place navigation

Spatial memory was assessed in a Morris water maze as previously reported (*Corradini et al, 2012*) with slight modification. Briefly, training and testing of mice were done under 12 lux diffuse light in a circular pool arena made of white polypropylene (diameter 100 cm, wall height 60 cm), filled with water (made opaque by the addition of milk) to a height of 30 cm and maintained at 24–26°C. A 10 cm diameter target platform of transparent polypropylene was placed 0.5 cm below the water surface in the N, S, E, or W quadrant at 20 cm from the wall. Each mouse was released to the pool from different starting points and was trained for a constant platform position over two days with five trials per day separated by 30–60 min intervals and each trial lasting maximally 120 s. The latency to find the escape platform was measured.

## T-maze

The T-maze was constructed of gray plexiglass, with stem length of 40 cm and arm length of 90 cm; each section was 11 cm wide with 19 cm high side walls. Animals were first food-deprived until reaching the 85% to 90% of their free-feeding body weight, and they were habituated to obtain food from cups placed at the ends of the arms of the T-maze for 5 days, with one acclimation per day (*Moy et al., 2008*). For each mouse, one arm was designated as the correct arm and a reinforcer (Kellogg's cereal) was available in the cup at the end of the arm each trial. The correct arm was on the right side for half of the mice, and on the left side for the other half, for each genotype. Each mouse was placed at the maze start and was given a free choice to enter either arm. Latency to enter an arm, number of errors in arm selection, and number of days to reach the criterion – that is, showing 70% of correct choices for three consecutive days - were recorded by a human observer. To assess the role of IL-1R pathway in the cognitive performances observed in IL-1R8 KO mice, an experimental group was treated with daily i.p. injection of 30 mg/kg Anakinra for the whole duration of the experiment. Animals showing freezing behavior, who did not move in the maze, were excluded from the analysis.

## In vitro experiments

### Cell cultures

Mouse hippocampal neurons were established from the hippocampi of embryonic stage E17 fetal mice (*Verderio et al., 1994*). The dissociated cells were plated onto glass coverslips coated with poly-L-lysine at densities of 95 cells/mm2. The cells were maintained in Neurobasal (Invitrogen, San Diego, CA) with B27 supplement and antibiotics, 2 mM glutamine, and glutamate (neuronal medium).

### Acute downregulation of MeCP2 expression

Silencing of MeCP2 was achieved via transfection of Sh CTRL and Sh MeCP2. Constructs are a kind gift of Michael Greenberg and were obtained as described in (*Zhou et al., 2006*).

### Transfection

Mouse hippocampal neurons were transfected with different plasmids using Lipofectamine 2000 (Invitrogen) at DIV12. Cultures were fixed and stained at DIV 16.

### Immunocytochemical staining and image analysis

Immunofluorescence staining was carried out as described in (*Verderio et al., 1994*) using the following antibodies: PSD-95 (1:400; monoclonal; UC Davis/NIH NeuroMab Facility, CA), MeCP2 (1:200; polyclonal; Cell Signaling), MAP2 (1:300; monoclonal; Immunological Sciences). Images were acquired using a Leica SPE confocal microscope. Images of primary hippocampal cultures were acquired with a Leica SPE confocal X 63 oil immersion lens (1,024 × 1,024 pixels). Each image

consisted of a stack of images taken through the z-plane of the cell. Confocal microscope settings were kept the same for all scans in each experiment.

### Dendritic spines analysis

For the analysis of spine density and size the program NeuronStudio was utilized. On the basis of morphology, the spines were classified into the following categories: (1) Thin: spines with a long neck and a visible small head; (2) Mushroom: big spines with a well-defined neck and a very voluminous head. PSD-95 puncta mean size was evaluated with Imaris, in particular with the Spots function. PSD-95 density analysis was performed using ImageJ software (NIH, Bethesda, Maryland, USA). At least four dendritic branches were analyzed for each neuron. The number of analyzed neurons is reported in each figure legend. At least three independent replications were performed for each experimental setting. Control data refer to each single experimental session.

## Cell culture electrophysiology

Whole cell voltage-clamp recordings were performed on wild type and transgenic embryonic hippocampal neurons maintained in culture for 13–15 DIV. During recordings cells were bathed in a standard external solution containing (in mM): 125 NaCl, 5 KCl, 1.2 MgSO4, 1.2 KH2PO4, 2 CaCl2, 6 glucose, and 25 HEPES-NaOH, pH 7.4. Recording pipettes were fabricated from borosilicate glass capillary using an horizontal puller (Sutter Instruments) inducing tip resistances of 3–5 MΩ and filled with a standard intracellular solution containing (in mM): 130 Cs-gluconate, 8 CsCl, 2 NaCl, 4 EGTA, 10 HEPES- NaOH, 2 MgCl2, 4 MgATP, and 0.3 Tris-GTP. For miniature AMPA-EPSC recordings tetrodotoxin 1µM, Bicuculline 20 µM and AP5 50 µM (Tocris) were added to standard extracellular solution to block the spontaneous action potentials propagation, GABA-A and NMDA receptors, respectively. Recordings were performed at room temperature in voltage clamp mode at holding potential of −70 mV using a Multiclamp 700B amplifier (Molecular Devices) and pClamp-10 software (Axon Instruments, Foster City, CA). Series resistance ranged from 10 to 20 MΩ and was monitored for consistency during recordings. Cells in culture with leak currents > 100 pA were excluded from the analysis. Signals were amplified, sampled at 10 kHz, filtered to 2 or 3 KHz, and analyzed using pClamp 10 data acquisition and analysis program. Electrophysiological mEPSC recordings of neurons WT, IL-1R8 KO and IL-1R8 KO IL-1R KO were always performed in the same experimental sessions.

Chemical Long Term Potentiation (LTP) was performed as in (*Menna et al., 2013*). Induction was performed stimulating synaptic NMDA receptors via glycine. For glycine-induced LTP experiments, hippocampal neurons were transfected at 10DIV with cDNA encoding for EGFP by using Lipofectamine 2000. After 6 days, cells were perfused with a solution containing (in mM) 125 NaCl, 5 KCl, 1.2 KH2PO4, 2 CaCl2, 1MgCl2, 6 glucose, and 25 HEPES-NaOH, TTX 0.001, Strychnine 0.001 and bicuculline methiodide 0.02 (pH 7.4, KRH) for 10 min then a solution devoid of Mg2+ and containing glycine (100 µM) was applied for 3 min followed by a wash and recovery in neuronal medium for at least 60 min. After 60 minutes cells were immediately fixed and stained. For patch clamp electrophysiology, the patch pipette electrode contained the following solution (in mM): 130 CsGluconate, 8 CsCl, 2 NaCl, 10 HEPES, 4 EGTA, 4 MgATP and 0.3 Tris-GTP.

## WB analysis

Samples containing 25 mg protein were resolved in 12% sodium dodecyl sulphate-polyacrylamide gels under reducing conditions. After transfer onto polyvinylidene diflouride membranes for 2 hr at 250 mA at 41C, blots were blocked for 1 hr at room temperature in a 5% MILK solution in phosphate-buffered saline (PBS) pH 7.4 and then incubated with PSD95 (1:10000; monoclonal; UC Davis/NIH NeuroMab Facility, CA), MeCP2 (1:1000; polyclonal; Cell Signaling), GAPDH (1:4000; polyclonal; Synaptic System, Goettingen, Germany) at 4°C overnight in PBS 0.5% Tween-20 (PBS-T). Subsequently, membranes were washed and incubated for 1 hr at room temperature in PBS-T with the secondary antibodies. Western blotting was performed by means of Chemi-Doc system + Image Lab software (Bio-Rad). Photographic development was by chemiluminescence (ECL, Amersham Bioscience or Immobilon substrate, Millipore). Western blot bands were quantified by the ImageJ program (rsb.info.nih.gov/ij).

## Transcriptomics and bioinformatic analyses

Total RNA from cortex from individual mice was isolated using TRI (#T9424; Sigma, Inc) and resuspended in 100 ul ddH2O treated with diethyl pyrocarbonate (DEPC). Samples were then incubated with DNase I (#79254; Qiagen, Inc) for 10 min at room temperature and precipitated with RNA grade potassium acetyate (Ambion, #AM9610). RNA pellets were finally resuspended in 20 ul ddH2O DEPC. Stranded mRNA-Seq multiplexed libraries were prepared from total RNA from mouse cortex following manufacturer's instructions (Illumina, Inc). To reduce biological variability, each library was performed with total RNA from three independent mice. A total of 3 libraries (9 mice in total) were performed per condition (12 libraries in total). Conditions were wild-type mice (C57bl/6j mice), wild-type mice treated with IL1$\beta$ 8 µg/kg o.n. (14 hr), IL-1R8 KO mice and IL-1R8 KO mice treated with Anakinra (30 mg/kg, i.p. administration) for three consecutive days. Sequencing was performed in a HiSeq 2500 apparatus in paired-end configuration (2 × 125 bp). To increase sequencing depth, samples were sequenced in two different lanes. All the libraries were loaded in each of the two lanes. Quality control of the raw data was performed with FastQC (http://www.bioinformatics.babraham.ac.uk/projects/fastqc/). Libraries were trimmed for adapter removal using Trimmomatic (*Bolger et al., 2014*) and mapped to reference genome (Ensembl GRCm38) using TopHat2 (*Kim et al., 2013*) and Bowtie2 (*Langmead et al., 2009*). Library sizes of primary mapped reads were between 70 and 96 million reads. Samtools was used to manipulate BAM files (*Li et al., 2009*). For calling of differentially expressed genes (DEG), mapped reads were counted with HTSeq v0.6.1 (*Anders et al., 2015*) and count tables were analysed using DeSeq2 v1.10.1 R-package (*Love et al., 2014*) with a design of one factor with four levels ('wild-type', 'wild-type + IL1$\beta$'', 'IL1-R8 KO', 'IL1-R8 KO + Anakinra') and differences between groups were tested using contrasts for wild-type + IL1$\beta$ versus wild-type; IL-1R8 KO versus wild-type; IL-1R8 KO + Anakinra versus wild-type. For consideration of differentially regulated genes between conditions, we used adjusted p-value<0.1 or adjusted p-value<0.05 as indicated in figure legends. Functional annotation and category and pathway analysis were carried out using WEB-based Gene SeT AnaLysis Toolkit (WebGestalt) (*Zhang et al., 2005*). All expression data are made publicly available in the GEO Series GSE80446.

## Statistical analysis

Data are presented as mean±standard error (SE) from the indicated number of experiments. Statistical analysis was performed using PRISM 6 software (GraphPad, Software Inc., San Diego,CA, USA). After testing whether data were normally distributed or not, the appropriate statistical test has been used. The Kolmogorov –Smirnov test was used to determine significance in cumulative distributions of mEPSC amplitudes and integrated density values. In particular, Mann-Whitney *t* test was used to determine significance in an average of mEPSC frequency. To determine significant differences in spine number we used Mann Whitney test or one-way ANOVA followed by specific multiple comparison post hoc tests (as indicated). The differences were considered to be significant if p≤0.05 and are indicated by (*) or (#); those at p≤0.01 are indicated by double (*) or (#); those at p≤0.005 are indicated by triple (*) or (#); those at p≤0.0001 by four (*) or (#).

## Acknowledgements

We thank Camilla Tafuro, Mattia Aime, Marilyn Scandaglia and Martina Molgora for help in some experiments. We are grateful to Dr. Michael Greenberg (Department of Microbiology and Molecular Genetics at Harvard Medical School) for shCTRL and shMeCP2 constructs. Work in our lab is supported by Cariplo 2012–0560, Ministero della Salute GR-2011–02347377, CARIPLO 2015–0594, to MM and by Cariplo 2015–0952 to RT. RT was previously supported by a Fondazione Umberto Veronesi Fellowship. IC is supported by a Fondazione Giancarla Vollaro Fellowship. JPLA is supported by grant SAF2014-60233-JIN from the Ministerio de Economía y Competitivdad (MINECO) co-financed by the European Regional Development Fund (ERDF). AB is supported by grants SAF2014-56197-R, SEV-2013-0317 and PCIN-2015-192-C02-01 from MINECO co-financed by the European Regional Development Fund (ERDF). AM acknolewledges a grant from Fondazione Cariplo (Contract n. 2015-0564)

# Additional information

## Funding

| Funder | Grant reference number | Author |
| --- | --- | --- |
| Fondazione Cariplo | 2015-0952 | Romana Tomasoni |
| Fondazione Cariplo | 2015–0952 | Romana Tomasoni |
| Ministerio de Economía y Competitividad | SAF2014-60233-JIN | Jose P Lopez-Atalaya |
| Fondazione Giancarla Vollaro | Fellowship | Irene Corradini |
| Fondazione Cariplo | 2015-0564 | Alberto Mantovani |
| Ministerio de Economía y Competitividad | SAF2014-56197-R | Angel Barco |
| Ministerio de Economía y Competitividad | SEV-2013-0317 | Angel Barco |
| Ministerio de Economía y Competitividad | PCIN-2015-192-C02-01 | Angel Barco |
| Fondazione Cariplo | 2015-0594 | Michela Matteoli |
| Ministero della Salute | GR-2011-02347377 | Michela Matteoli |
| Fondazione Cariplo | 2012-0560 | Michela Matteoli |

The funders had no role in study design, data collection and interpretation, or the decision to submit the work for publication.

## Author contributions

RT, Conceptualization, Data curation, Formal analysis, Investigation, Writing—original draft, Writing—review and editing; RM, IC, AC, MR, CM, DP, Data curation, Formal analysis, Investigation; JPL-A, Conceptualization, Data curation, Formal analysis; CG, AM, Conceptualization, Writing—original draft, Writing—review and editing; EM, Conceptualization, Data curation, Formal analysis, Writing—original draft, Writing—review and editing; AB, Data curation, Formal analysis, Writing—original draft; MM, Conceptualization, Data curation, Supervision, Funding acquisition, Writing—original draft, Project administration, Writing—review and editing

## Author ORCIDs

Romana Tomasoni, http://orcid.org/0000-0002-3374-2698
Michela Matteoli, http://orcid.org/0000-0002-3569-7843

## Ethics

Animal experimentation: All the experimental procedures followed the guidelines established by the European Legislation (Directive 2010/63/EU), and the Italian Legislation (L.D. no 26/2014).

# Additional files

## Supplementary files

• Supplementary file 1. Differentially expressed mRNAs in the hippocampus of IL-1$\beta$ treated mice

• Supplementary file 2. Differentially expressed mRNAs in the hippocampus of IL-1R8 (aka TIR8) KO mice

• Supplementary file 3. Genes differentially expressed in the hippocampus of both IL-1$\beta$ treated mice and IL-1R8 (aka TIR8) KO.

- Supplementary file 4. (A) Webgestalt GO-Biological process enrichment analysis for genes differentially expressed after treatment with IL-1$\beta$ (see *Supplementary file 1*). (B) Webgestalt GO-Biological process enrichment analysis for genes differentially expressed in IL-1R8 KO mice (see *Supplementary file 2*). (C) Full Webgestalt GO enrichment analysis (BP, MF and CC) for genes differentially expressed both in IL-1R8 KO mice and in IL-1$\beta$ treated mice (see *Supplementary file 3*).

- Supplementary file 5. Raw data of all the values used for graphics in the different panels.

- Supplementary file 6. Statistic analysis of the graphics in the different panels.

### Major datasets

The following dataset was generated:

| Author(s) | Year | Dataset title | Dataset URL | Database, license, and accessibility information |
|---|---|---|---|---|
| Romana Tomasoni, Jose P Lopez-Atalaya, Angel Barco, Michela Matteoli | 2017 | Quantitative Analysis of cortical transcriptomes through Next Generation Sequencing from wild-type mice, wild-type mice treated with IL1b, IL-1R8 KO mice and IL-1R8 KO mice treated with IL1b antagonist Anakinra | https://www.ncbi.nlm.nih.gov/geo/query/acc.cgi?acc=GSE80446 | Publicly available at the NCBI Gene Expression Omnibus (accession no: GSE80446) |

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
