## [Decision Letter]

[Editors’ note: this article was originally rejected after discussions between the reviewers, but the authors were invited to resubmit after an appeal against the decision.]

Thank you for submitting your work entitled "Deregulated IL-1 Receptor Pathway Increases MeCP2 in Neurons and Induces MeCP2- Dependent Synaptic Defects" for consideration by *eLife*. Your article has been favorably evaluated by a Senior Editor and three reviewers, one of whom, Yukiko Goda (Reviewer #1), is a member of our Board of Reviewing Editors.

Our decision has been reached after consultation between the reviewers. Based on these discussions and the individual reviews below, we regret to inform you that your work will not be considered further for publication in *eLife*, at least in its present form.

While all three reviewers felt that the findings are interesting and novel, some of the data are not sufficiently compelling to support the major conclusions. Given that an extensive revision would be required to fully address the specific concerns raised by the reviewers, we are returning the manuscript to you. However, we encourage you to resubmit the work as a new manuscript if you can provide data that strongly support the hypothesis.

*Reviewer #1:*

Here Tomasoni et al. explore the relationship between inflammation and brain function by focusing on IL-1R8 and characterizing the defects in synaptic functions in IL-1R8 KO neurons. The observed alterations in spine morphology, density and LTP in IL-1R8 KO neurons could be restored by interfering with IL-1R or by inhibiting mTOR/PI3K/Akt pathway. Similarly, downregulation of MeCP2 can reverse the synaptic defects in IL-1R8 KO neurons, suggesting that IL-1R signaling pathways controls the level of MeCP2 expression to affect synapse function and cognition. The authors also show that impaired cognitive performance of in IL-1R8 KO mice could be attenuated by blocking IL-1 activity as tested for the novel place object recognition test. Altogether, the topic is timely and the main findings are interesting. However, in the present manuscript, several key controls are missing, which make it difficult to fully interpret the data as shown. The interaction between IL-1R8 and IL-1R1, and how their signaling is related to MeCP2 expression level needs to be better characterized.

1) Figure 3: Interestingly, in panel B, treatment of wild type neurons with IL-1Ra blocks the increase in mushroom spines associated with LTP induction, such that IL-1R could apparently act positively in supporting LTP when IL-1R8 expression is not perturbed. To fully interpret this result, images should be shown in panel A, and the control or LTP induction in wild type neurons treated with IL1-Ra should be included for PSD95 puncta and mEPSC amplitude and frequency analysis.

2) Figure 4: Following from the point above, it would be helpful to compare also how the transcriptome landscape changes in wild type animals upon treatment with IL-1Ra (Anakinra).

3) Figure 5: Rather than simply reversing the decrease in the density of mushroom spines and PSD95 puncta to wild type controls, treatment of IL-1R8 KO neurons with inhibitors of mTOR/PI3K/Akt signaling promotes a robust increase in these parameters. How do such effects compare to the effects of these inhibitors on wild type neurons? Do they interact with the blockade of IL-1R signaling with Anakinra, which prevents the LTP-dependent increase in mushroom spines?

4) Figure 6: Does the rescue of LTP-associated increase in mushroom spines, PSD-95 puncta, and mEPSC amplitude and frequency by knocking down MeCP2 in IL-1R8 KO neurons occlude the similar effects of IL-1Ra treatment also observed in IL-1R8 KO neurons? If MeCP2 is downstream to IL-1R, then one would expect for occlusion and this should be tested.

5) Was the MeCP2 transcript level changed in IL-1R8 KO mice and in WT mice treated with IL-1beta?

6) Figure 7: It would be crucial to test whether Anakira treatment in WT mice affects hippocampal spatial memory as a control. In order to claim for the rescue of cognitive deficits in IL-1R8 KO mice by blocking IL-1R signaling more compellingly, the mice should be subjected to additional behavior paradigms with or without treatment with blockers.

*Reviewer #2:*

The current manuscript by Tomasoni et al., examines the molecular mechanisms by which IL-1R8 deficiency results in brain deficits. Through a range of genetic and pharmacological manipulations, the authors present evidence that deficiency in IL-1R8 alters spine number, mEPSC frequency and chemical LTP-induced plasticity through interleukin 1 hyper activation. However, some key data is not convincing raising questions about some of the conclusions. Although the link between IL-1R activation and MeCP2 is novel, the relevance of this finding is not supported by the data. Furthermore, there is no clear rationale for some of the experiments presented. Given these concerns, the manuscript should not be accepted in its current form.

1) Many of the images of neurons are not convincing. For example, the section of the dendrite shown under control and experimental conditions seem to come from different part of the dendritic arbour as the thickness is different (i.e. Figure 1, Figure 3, Figure 4, Figure 5, Figure 7, Figure 8). In addition, the majority of the images do not match the corresponding quantification.

2) Figure 1: PSD95 staining intensity is presented. It would be interesting to know the size of spines given that this is proportional to the PSD area. Furthermore, the decrease in PSD95 puncta intensity should correlate with a decrease in mEPSC amplitude, which is not seen in this experiment. The authors should provide an explanation for this.

3) In Figure 2, the authors examined vGlut1 (presynaptic marker) area in the IL1R8-/- mouse. These images are extremely poor and unclear. They do not seem to come from a confocal image.

4) Figure 1: The traces of electrophysiological recordings do not reflect the graphs presented. For example, Figure 1 shows a frequency of about 6Hz (based on the number of events depicted in the traces) yet the graph indicates a frequency of 1.2Hz.

5) The amplitude of EPSCs is not homogenously presented. For example, Figure 1 shows a cumulative probability graph, Figure 3, Figure 5 and Figure 8 are normalised (but not stated as such) and Figure 4 shows raw values. Consistency is required.

6) Figure 4: Pharmacological or genetic rescue experiments show an overshooting effect (increase above control condition) see Figure 4 and E. Does it mean that the IL-1R -/- has an effect on its own?

7) The transcriptome analyses were done in cortical neurons but all the conclusions are based on the hippocampus. The data should be taken with caution.

8) The authors use several drugs (like rapamycin, LY or Anakinra) to rescue the phenotypes but they don't show the effect of these drugs alone preventing solid conclusion.

9) A rescue of the Mecp2 shRNA is required. To conclude that Mecp2 is involved in the in vivo phenotype of IL-1R8-/- mice, the authors should knock-down Mecp2 and perform behaviour experiments.

10) As the authors propose that defects in IL1-R8 deficient mice are due to hyperactivity of IL pathway, it would be interesting to examine IL1 level in this KO mouse.

*Reviewer #3:*

This study by Tomasoni et al. reports the role of IL-1 receptor pathway in the regulation of excitatory synapses through MeCP2 regulation. In support of this conclusion, the authors provide evidence supporting that IL-1R8 antagonizes the IL-1R pathway, which negatively regulates excitatory synapse density and LTP through upregulation of MeCP2. The authors use various approaches including dissociated neuron culture, immunofluorescence staining, electrophysiology, pharmacology, and knockdown. This manuscript is a comprehensive in the experimental designs, and the data are largely convincing. Particularly novel findings seem to include that IL-1R8 negatively regulates IL-1R to promote excitatory synapses, and that MeCP2 negatively regulates excitatory synapses by acting downstream of the IL-1R-mTOR pathway.

1) Figure 7 shows that the mTOR pathway acts downstream of IL-1R8. Are the protein levels or activity (phosphorylation) of mTOR pathway proteins increased in the IL-1R8-mutant mice? If so, is the mTOR pathway suppressed in anakinra-treated IL-1R-mutant mice?

[Editors’ note: what now follows is the decision letter after the authors submitted for further consideration.]

Thank you for resubmitting your work entitled "Lack of IL-1R8 in neurons causes hyperactivation of IL-1 receptor pathway and induces MECP2-dependent synaptic defects" for further consideration at *eLife*. Your revised article has been favorably evaluated by Jonathan Cooper (Senior Editor) and the Reviewing Editor.

The manuscript has been significantly improved with new experiments and extensive editing of the text. However, one remaining issue needs to be addressed before acceptance, as outlined below:

Figure 7. Why are immunofluorescence signals for MAP2 (panel C) and beta3 tubulin (panel D) considerably brighter in IL-1R8KO/treatment conditions compared to WT? This raises concerns about the quantification and comparison of MeCP2 signal intensity across conditions if images for different experimental conditions have been captured under different settings.

---

## [Author Response]

[Editors’ note: the author responses to the first round of peer review follow.]

*Reviewer #1:*

*[…] 1) Figure 3: Interestingly, in panel B, treatment of wild type neurons with IL-1Ra blocks the increase in mushroom spines associated with LTP induction, such that IL-1R could apparently act positively in supporting LTP when IL-1R8 expression is not perturbed. To fully interpret this result, images should be shown in panel A, and the control or LTP induction in wild type neurons treated with IL1-Ra should be included for PSD95 puncta and mEPSC amplitude and frequency analysis.*

To address whether IL-1R acts positively in supporting LTP when IL-1R8 expression is not perturbed, wild type neurons were treated with IL1-Ra and examined for mushroom density, PSD95 puncta and mEPSC amplitude and frequency with or without LTP, as required by the reviewer. The results indicate that treated neurons display increased PSD95-positive puncta density as well as increased mEPSC amplitude and frequency. No potentiation of any of these parameters occurs after application of the LTP protocol, indicating inability to undergo plasticity. All these results are now included in the new Figure 4 (panels A-F). Furthermore, in line with these observations, neurons genetically devoid of the IL-1 receptor (IL-1R), similarly to wt neurons treated with IL-1Ra, were unable to undergo plasticity phenomena, as indicated by the lack of increase of either mushroom spines after LTP induction (results added as numerical data in the subsection “Defects in structure and function of IL-1R8 KO neurons are reversed by blocking IL-1 receptor activity”). Therefore, either pharmacological or genetic silencing of IL-1R is per sesufficient to alter dendritic spine morphology and plasticity. These data confirm previous literature evidence showing that blocking IL-1 receptors with IL-1ra impairs the maintenance of LTP (Schneider et al., 1998, Coogan et al., 1999; Avital et al., 2003; Costello et al. 2011). Images of WT neurons have been added to panel A of new Figure 4, as required by the reviewer.

*2) Figure 4: Following from the point above, it would be helpful to compare also how the transcriptome landscape changes in wild type animals upon treatment with IL-1Ra (Anakinra).*

We have collected the material and sent it to the company for analysis. The results are not back yet and, however, they will need to be deeply analyzed, thus requiring few more months before being available. However, based on the functional evidences obtained in wt neurons exposed to IL-1Ra (new Figure 4) and in IL-1R KO neurons (numerical data added in the subsection “Defects in structure and function of IL-1R8 KO neurons are reversed by blocking IL-1 receptor activity”) and in consideration of the behavioral defects of wt mice treated with anakinra, which we have now added to the revised manuscript (new Figure 10), we can easily predict that the treatment with IL-1Ra will change the transcriptome landscape in wt mice. We would like to focus on this specific issue in a future research, given that a new study will be anyway necessary to provide a full description of the genes found to be changed upon IL-1R hyperactivation (reported in Figure 5).

*3) Figure 5: Rather than simply reversing the decrease in the density of mushroom spines and PSD95 puncta to wild type controls, treatment of IL-1R8 KO neurons with inhibitors of mTOR/PI3K/Akt signaling promotes a robust increase in these parameters. How do such effects compare to the effects of these inhibitors on wild type neurons? Do they interact with the blockade of IL-1R signaling with Anakinra, which prevents the LTP-dependent increase in mushroom spines?*

We have now added to the manuscript a novel figure illustrating the density of mushroom spines and postsynaptic puncta in wt neurons treated with inhibitors of mTOR/PI3K/Akt (new Figure 6). The results indicate that wt neurons exposed to the different inhibitors display an increase in spine and postsynaptic marker density. These data reveal that endogenous activation of mTOR/PI3K/Akt pathways is required for correct spine morphogenesis. This consideration has been added to the subsection “L-1R effects on hippocampal synapses are mediated by the PI3K/AKT/mTOR pathway”. This is in agreement with the evidence that physiological activation of IL-1R is required for spine morphogenesis and plasticity (see answer to point 1) and may explain why, rather than simply reversing the decrease in the density of mushroom spines and postsynaptic puncta to wild type controls, treatment of IL-1R8 KO neurons with inhibitors of mTOR/PI3K/Akt signaling promotes an increase in these parameters. The very poor health of neurons exposed to IL-1Ra together with inhibitors of mTOR/PI3K/Akt pathway hampered the possibility to examine the effects of their simultaneous presence.

*4) Figure 6: Does the rescue of LTP-associated increase in mushroom spines, PSD-95 puncta, and mEPSC amplitude and frequency by knocking down MeCP2 in IL-1R8 KO neurons occlude the similar effects of IL-1Ra treatment also observed in IL-1R8 KO neurons? If MeCP2 is downstream to IL-1R, then one would expect for occlusion and this should be tested.*

We performed the experiments required by the reviewer and found that, by applying the two treatments together, LTP was only partially restored, at difference with neurons where only IL-1Ra was applied. See Figure 11.

Author response image 1.**DOI:**
http://dx.doi.org/10.7554/eLife.21735.020

However, given that the results of this experiment are not easy to interpret, we attempted to provide a more direct answer to the reviewer’s question of whether MeCP2 is downstream to IL-1R, by pharmacologically or genetically silencing IL-1R and quantifying MeCP2 expression. We show in the revised manuscript that application of IL-1Ra to wt neurons results in a reduction of MeCP2 levels (new Figure 7); also MeCP2 levels are significantly lower in both hippocampus and cortex of mice lacking IL-1R KO, compared to wt (results of WB quantification added as numerical values in the text, subsection “The transcriptional regulator MeCP2 mediates the alterations in spine morphogenesis, synaptic transmission and synaptic plasticity observed in IL-1R8 KO hippocampal neurons”). These data further support the view that MeCP2 is downstream to IL-1R.

*5) Was the MeCP2 transcript level changed in IL-1R8 KO mice and in WT mice treated with IL-1beta?*

The MeCP2 transcript levels are not changed in IL-1R8 KO mice, suggesting that the effects on MeCP2 occur at translational and not transcriptional level. This information, including the numerical data showing that, is now reported in the revised manuscript (subsection “The transcriptional regulator MeCP2 mediates the alterations in spine morphogenesis, synaptic transmission and synaptic plasticity observed in IL-1R8 KO hippocampal neurons”).

*6) Figure 7: It would be crucial to test whether Anakira treatment in WT mice affects hippocampal spatial memory as a control. In order to claim for the rescue of cognitive deficits in IL-1R8 KO mice by blocking IL-1R signaling more compellingly, the mice should be subjected to additional behavior paradigms with or without treatment with blockers.*

To address the reviewer’s request, we tested IL-1R8 KO mice for the Morris water maze and the rewarded T maze. In line with the concept that alternation, either rewarded or spontaneous, detects hippocampal dysfunction even better than the Morris water maze (reviewed in Deacon et al., 2006), no significant defects of IL-1R8 KO mice in the Morris test were apparent, while an impairment of spatial memory was detected in the rewarded T-maze task. Treatment with Anakinra ameliorated the performance. A novel figure (Figure 10) has been produced with a complete behavioral characterization. The new figure also includes the effect of Anakinra in wt mice tested for novel place object recognition, as required by the reviewer. The description of these results is included subsection “Inhibition of IL-1β receptor by Anakinra rescues behavioral defects” of the Results section.

*Reviewer #2:*

*The current manuscript by Tomasoni et al., examines the molecular mechanisms by which IL-1R8 deficiency results in brain deficits. Through a range of genetic and pharmacological manipulations, the authors present evidence that deficiency in IL-1R8 alters spine number, mEPSC frequency and chemical LTP-induced plasticity through interleukin 1 hyper activation. However, some key data is not convincing raising questions about some of the conclusions. Although the link between IL-1R activation and MeCP2 is novel, the relevance of this finding is not supported by the data. Furthermore, there is no clear rationale for some of the experiments presented. Given these concerns, the manuscript should not be accepted in its current form.*

In his/her comments, the reviewer claims that “some key data are not convincing and this raises questions about some of the conclusions of our manuscript”. While this appears like a quite serious general concern, we noticed that, actually, most of the reviewer’s points relate with figure panels used as representative images. We apologize with the reviewer for the choice of our images and we provide novel panels to address his/her concerns.

*1) Many of the images of neurons are not convincing. For example, the section of the dendrite shown under control and experimental conditions seem to come from different part of the dendritic arbour as the thickness is different (i.e. Figure 1, Figure 3, Figure 4, Figure 5, Figure 7, Figure 8). In addition, the majority of the images do not match the corresponding quantification.*

We apologize with the reviewer for the not convincing images. We believe that this is partly due to the fact that the pdf generated by the system and sent to the reviewers is of low quality and we would ask the Editorial Office to please send the figures to the reviewers in a different format, if possible. However, in order to answer the reviewer’s criticism, we have now changed most of the images of the revised manuscript, choosing more homogeneous fields, with a similar thickness of the dendritic arbour and paying more attention in choosing fields which match better the corresponding quantification. The revised figures which contain all or most novel panels are the following: Figure 1, Figure 3, Figure 4, Figure 4, Figure 6.

*2) Figure 1: PSD95 staining intensity is presented. It would be interesting to know the size of spines given that this is proportional to the PSD area. Furthermore, the decrease in PSD95 puncta intensity should correlate with a decrease in mEPSC amplitude, which is not seen in this experiment. The authors should provide an explanation for this.*

We wish to point out that the PSD-95 staining intensity quantitation in original Figure 1 referred to the WB analysis of wt and IL-1R8 KO cultured neurons. Concerning PSD-95 puncta, our original manuscript reported only their density (number of puncta/micron length of parent dendrite) assessed by confocal analysis. To meet the reviewer’s request, we are now providing a quantitation of spine width and PSD-95 puncta area, both of which are significantly reduced in IL- 1R8 KO neurons (new Figure 1, subsection “Altered synaptic architecture and function in IL-1R8 KO hippocampal neurons” of Results). Concerning the reviewer’s statement that “the decrease in PSD95 puncta intensity should correlate with a decrease in mEPSC amplitude”, several reports in literature indicate that, in fact, mEPSC frequency, and not amplitude, is impacted by tuning the levels of postsynaptic proteins, like PSD-95, PSD-93, SAP102 or GluA2 (Sun and Turrigiano, J Neurosci 2011; Saglietti et al. Neuron 2007). In particular, Sun and Turrigiano have shown that overexpression of PSD-95 increases mEPSC frequency, but not amplitude, and that knockdown of PSD-95 does not affect mEPSC amplitude. Our data are therefore in line with already published evidence.

*3) In Figure 2, the authors examined vGlut1 (presynaptic marker) area in the IL1R8-/- mouse. These images are extremely poor and unclear. They do not seem to come from a confocal image.*

We apologize with the reviewer for the poorness of the images in Figure 2. As mentioned before, we believe that this may be, at least in part, the consequence of the bad quality of the pdf file generated by the system. However, we are now providing a better quality, alternative image for Figure 2.

*4) Figure 1: The traces of electrophysiological recordings do not reflect the graphs presented. For example, Figure 1 shows a frequency of about 6Hz (based on the number of events depicted in the traces) yet the graph indicates a frequency of 1.2Hz.*

Following the reviewer’s notice, we have realized that in fact the traces we have provided do not reflect the exact average frequency reported by the quantitation. While we point out that electrophysiological traces are taken as representative and are meant to make a qualitative point, which is then quantified in the histograms, we have now substituted the electrophysiological traces, choosing examples which display the exact average frequency emerging from the quantitative analysis (see new Figure 1, old Figure 1).

*5) The amplitude of EPSCs is not homogenously presented. For example, Figure 1 shows a cumulative probability graph, Figure 3, Figure 5 and Figure 8 are normalised (but not stated as such) and Figure 4 shows raw values. Consistency is required.*

We now show both cumulative and normalized amplitudes in Figure 1. In the case of LTP, in order to make the figure more comprehensible to the readers, only the normalized frequency and amplitude are shown, as usually displayed in literature (Lu et al., 2001; Fortin et al., 2010). We have now specified in the figure legend and on the y axis when normalization was performed (Figure 3; Figure 4).

*6) Figure 4: Pharmacological or genetic rescue experiments show an overshooting effect (increase above control condition) see Figure 4 and E. Does it mean that the IL-1R -/- has an effect on its own?*

We thank the reviewer for raising this point, which has now been addressed. The revised version of the manuscript reports the effects of all pharmacological treatments (IL-1Ra and all the inhibitors of mTOR/PI3K/Akt signaling) in wt neurons (see new Figure 4 and new Figure 6). These data indicate that pharmacological blockade of either IL-1R or mTOR/PI3K/Akt pathway is per se sufficient to alter dendritic spine morphology and plasticity (sentences added in the subsections “Defects in structure and function of IL-1R8 KO neurons are reversed by blocking IL-1 receptor activity” and “IL-1β treatment and IL-1R8 deficiency trigger overlapping gene programs related to hippocampal development and synaptic transmission”). See also answers to points 1 and 3 of reviewer 1.

*7) The transcriptome analyses were done in cortical neurons but all the conclusions are based on the hippocampus. The data should be taken with caution.*

We thank the reviewer for his/her note of caution. However, we wish to point out that the WB analysis of MeCP2 levels was originally performed also in the cortex in IL-1R8 KO mice, with or without IL-1Ra treatment, with similar results (Figure 9 of the original manuscript, now Figure 9). In order to further meet the reviewer’s point, we added to the revised manuscript the confocal quantitation of MeCP2 levels in mice cortex and corresponding images (new Figure 9).

*8) The authors use several drugs (like rapamycin, LY or Anakinra) to rescue the phenotypes but they don't show the effect of these drugs alone preventing solid conclusion.*

The revised version of the manuscript reports the quantitation of the effects of all drugs in wt neurons (new Figure 4 for IL-1Ra and new Figure 6 for LY, wortmannin and rapamycin).

*9) A rescue of the Mecp2 shRNA is required. To conclude that Mecp2 is involved in the* in vivo *phenotype of IL-1R8-/- mice, the authors should knock-down Mecp2 and perform behaviour experiments.*

In our manuscript, we have shown that MeCP2 is involved in the neuronal synaptic phenotype, while we have never claimed that MeCP2 is involved in the in vivo, behavioral phenotype. Obtaining a direct evidence of this would be in fact very challenging. The viral siRNA intrahippocampal injection allows access only to the dorsal hippocampus leaving the ventral one untouched. Due to the limited virus spreading upon intraparenchymal injection, a relatively low number of cells is transduced, thus failing to achieve a sufficient gene correction for behavioral studies (Parr-Brownlie et al., Front Mol Neurosci 2015). We already experienced this limitation in our previous studies (Fossati et al., Cell Death and Differentiation 2015). Also, MeCP2 is widely involved in several neuronal functions, and even small variations of MeCP2 results per se in robust behavioral deficits (see for example Chao & Zoghbi Nature Neuroscience, 2012; Samaco et al., 2013), implying that only physiological levels of MeCP2 can guarantee the proper brain functioning. Finely controlling levels of MeCP2 expression in vivo using the above strategy is not attainable. In order to avoid possible misunderstandings and make clear that we cannot conclude that Mecp2 is involved in the in vivo phenotype of IL-1R8KO mice, we have now reorganized the in vivo section of our manuscript and smoothened our conclusions further (Results and Discussion).

*10) As the authors propose that defects in IL1-R8 deficient mice are due to hyperactivity of IL pathway, it would be interesting to examine IL1 level in this KO mouse.*

Costello et al., in their 2011 paper, have already shown that IL-1alpha but not IL-1beta is increased in IL1-R8 deficient mice.

Reviewer #3:

*[…] 1) Figure 7 shows that the mTOR pathway acts downstream of IL-1R8. Are the protein levels or activity (phosphorylation) of mTOR pathway proteins increased in the IL-1R8-mutant mice? If so, is the mTOR pathway suppressed in anakinra-treated IL-1R-mutant mice?*

From our transcriptomic analysis we know that RNA levels for mTOR are not changed in IL-1R8 KO mice brain (see Figure 12).

Author response image 2.**DOI:**
http://dx.doi.org/10.7554/eLife.21735.021

This suggests that is the activity of the pathway likely increased in IL-1R8 KO neurons, as already reported in IL-1R8 KO T cells (Gulen et al. 2010). Consistently, rapamycin rescues the phenotypes, indicating the increased activity of the pathway in our experimental conditions.

[Editors’ note: the author responses to the re-review follow.]

*The manuscript has been significantly improved with new experiments and extensive editing of the text. However, one remaining issue needs to be addressed before acceptance, as outlined below:*

*Figure 7. Why are immunofluorescence signals for MAP2 (panel C) and beta3 tubulin (panel D) considerably brighter in IL-1R8KO/treatment conditions compared to WT? This raises concerns about the quantification and comparison of MeCP2 signal intensity across conditions if images for different experimental conditions have been captured under different settings.*

In order to quantify MeCP2 levels of expression, confocal images were captured using settings which were maintained strictly constant among the different experimental conditions. Sometimes, the focal plan of microtubules (stained for MAP2 or beta3 tubulin in order to provide a general picture of neuronal cells) was not coincident with the optimal focal plan used for capturing MeCP2-stained nuclei, thus resulting in apparent differences in the staining intensity of microtubules. Although the staining for microtubules is not quantified, thus excluding any possible impact on MeCP2 analysis, we agree that this inhomogeneity among different pictures may result misleading. We are grateful to the Editors for noticing this point. We have now substituted panels in Figure 7 in order to have more homogeneous pictures.